# Early-mid Miocene erosion rates inferred from pre-Dead Sea rift Hazeva River fluvial chert pebbles using cosmogenic $^{21}$Ne

Michal Ben-Israel[1], Ari Matmon[1], Alan J. Hidy[2], Yoav Avni[3], Greg Balco[4]

[1]*The Institute of Earth Sciences, Hebrew University of Jerusalem, Jerusalem, 91904, Israel*

[2]*Center for Accelerator Mass Spectrometry, Lawrence Livermore National Laboratory, Livermore, CA 94550, USA*

[3]*Geological Survey of Israel, Yesha'yahu Leibowitz 32, Jerusalem, 96921 Israel*

[4]*Berkeley Geochronology Center, Berkeley, California 94709, USA*

*Correspondence to*: Michal Ben-Israel (michal.benisrael@mail.huji.ac.il)

**Abstract.** In this work, we utilize a novel application of cosmogenic $^{21}$Ne measurements in chert
to compare exposure times measured in eroding surfaces in the Jordanian Central Plateau with exposure times from chert pebbles transported by the Miocene Hazeva River. The Miocene Hazeva River was a large fluvial system (estimated catchment size >100,000 km$^2$) that drained the Arabian Plateau and Sinai Peninsula into the Mediterranean Sea during the early-mid Miocene. It was established after the rifting of the Red Sea uplifted the Arabian Plateau during the Oligocene.
Following late Miocene to early Pliocene subsidence along the Dead Sea Rift, the Hazeva drainage system was abandoned and dissected, resulting in new drainage divides on either side of the rift. We find that modern erosion rates derived from cosmogenic $^{21}$Ne, $^{26}$Al, and $^{10}$Be in exposed *in situ* chert nodules to be extremely slow, between 2-4 mm/kyr. Comparison between modern and paleo-erosion rates, measured in chert pebbles, is not straightforward, as cosmogenic $^{21}$Ne was
acquired partly during bedrock erosion and partly during transport of these pebbles in the Hazeva River. However, $^{21}$Ne exposure times calculated in Miocene cherts are generally shorter (range between $0^{+59}_{-0}$ and 242±113 kyr) compared to exposure times calculated in the currently eroding chert nodules presented here (269±49 and 378±76 kyr) and other chert surfaces currently eroding

in hyperarid environments. Miocene exposure times are shorter even when considering that they account for bedrock erosion in addition to maintained transport along this large river. Shorter exposure times in Miocene cherts correspond to faster paleo-erosion rates, which we attribute to a combination of continuous surface uplift and significantly wetter climatic conditions during the early-mid Miocene.

## 1. Introduction

Tectonic and climatic conditions control geomorphological processes through surface uplift, rock weathering, and sediment generation and transport (e.g., Allen, 2008; Whipple, 2009; Whittaker, 2012). Changes in rates of continental uplift and climatic conditions control rates of erosion control sediment production, transport, and storage and influence fluvial systems and their associated sediment archives (e.g., DiBiase and Whipple, 2011; Ferrier et al., 2013; Vance et al., 2003). Cosmogenic nuclides, mostly radiogenic $^{26}$Al and $^{10}$Be, have been used extensively to study weathering and erosion rates in fluvial systems across different scales and geological settings (e.g., Bierman, 1994; von Blanckenburg, 2005). The decreased preservation of older sediments in fluvial systems, due to burial or recycling, adds difficulty to the reconstruction of past tectonic or climatic conditions with increased sediment age (e.g., Anderson et al., 1996; Guralnik et al., 2011; Schaller et al., 2002). Furthermore, even when geological circumstances do allow for the preservation of older sediments, rates prior to the Pliocene cannot be quantified with the more commonly used cosmogenic radionuclides ($^{10}$Be and $^{26}$Al) due to their half-lives (1.38 Myr and 716 kyr, accordingly; Ivy-Ochs and Kober, 2008). Unlike their radioactive counterparts, stable cosmogenic nuclides have the potential to quantify rates of surface processes as far back as Lower Cretaceous (Balco et al., 2019; Ben-Israel et al., 2018; Dunai et al., 2005; Libarkin et al., 2002; Sinclair et al., 2019). Here, we apply stable cosmogenic $^{21}$Ne to sediments deposited during the early-mid Miocene (~18 Ma) by the Hazeva River. This massive fluvial system drained parts of the Arabian Peninsula and Sinai into the Mediterranean prior to the subsidence of the Arava Valley along the Dead Sea transform (Garfunkel and Horowitz, 1966; Zilberman and Calvo, 2013). We quantify the time of exposure during erosion and transport of Miocene chert pebbles deposited by the Hazeva River and compare it to exposure times of chert that has been eroding over the recent past (~$10^5$ yr). Through this comparison, we quantify differences between erosion rates during early-

mid Miocene and rates of hyperarid environments eroding today, and examine the possible influence of the tectonic and climatic conditions that operated in the region during this time.

## 2. Geological Setting

Following an extended period of transgression that ended in the late Eocene, the Mediterranean Sea retreated to its current location (Garfunkel and Horowitz, 1966). This period of relative tectonic tranquility was followed by a series of tectonic and magmatic events that resulted in the rifting of the Red Sea and the Gulf of Aden in the late Eocene to early Oligocene (~35-30 Ma; e.g., Bohannon et al., 1989; Bosworth et al., 2005; Omar and Steckler, 1995). During the last 20-30 Myr, regional doming associated with the emergence of the Afar plume uplifted the Arabian Peninsula from near sea level to its present elevation of ~1km (e.g., Feinstein et al., 2013; Morag et al., 2019; Wilson et al., 2014). As a result of this uplift, widespread denudation followed, and a regional truncation surface developed in the northern Red Sea and the southern Levant exposing older strata down to Precambrian formations depending on the preexisting structure (Avni et al., 2012). Following these events, during the early-mid Miocene, the uplifted region was drained by a newly established fluvial system, termed the Hazeva River, which flowed northwestward from the eroded terrains towards the Mediterranean Sea, and drained an estimated area >100,000 km$^2$ (Garfunkel and Horowitz, 1966; Zilberman and Calvo, 2013; Fig. 1). The Hazeva fluvial system operated until the subsidence of the Dead Sea Rift, during the late Miocene to early Pliocene, brought on a dramatic change in morphology, which led to the disruption of this massive fluvial system, the last of its kind in the region (Garfunkel, 1981). By the early Pliocene, new independent drainage systems replaced the Hazeva River, draining the region toward the Dead Sea Basin (Avni et al., 2001).

At present, the mostly clastic sedimentary Miocene sequence deposited by the Hazeva River is preserved mainly in structural lows, karstic systems, and abandoned stream valleys in southern Israel, eastern Sinai, and Jordan (Calvo and Bartov, 2001; Fig. 2). The sediments associated with this Miocene fluvial system comprise the upper section of the Hazeva formation in southern Israel. This formation is divided into two major parts, the lower includes autochthonous conglomerates and lacustrine carbonate units, and the upper part is comprised of allochthonous clastic sequences typical of fluvial environments (Calvo, 2002). Here, we focus on the allochthonous upper part of the Hazeva formation and examine two different silicate members eroded from the uplifted

Arabian Plateau and Sinai and deposited simultaneously by the Hazeva River (Zilberman and Calvo, 2013). The first member is sub-rounded monocrystalline quartz-arenite, eroded from Phanerozoic Nubian sandstone, as well as from outcrops of Precambrian crystalline rocks of the Arabian-Nubian shield (Calvo and Bartov, 2001). The second member consists of well-rounded chert pebbles, either interbedded with the quartz sand or forming horizons of pebbles in the sandy sequence (Zilberman and Calvo, 2013). The chert comprising these pebbles is sourced only from east of the Dead Sea Rift, and therefore fluvial deposits on the west side containing this "imported chert" (Kolodny, 1965) must have been emplaced prior to rifting. The onset of the Hazeva River is constrained by the Karak dike (~20 Myr) which intrudes the lower section of the Hazeva formation (Calvo and Bartov, 2001). During the Miocene climatic conditions in the Levant are hypothesized to have been wetter (e.g., Kolodny et al., 2009). Currently, this region is part of a middle latitude dry warm desert extending from northern Africa to western Asia, with the Negev Desert remaining hyperarid at least since the middle Pleistocene (Amit et al., 2006).

## 3. Methodology and Analytical Procedures

### 3.1 Sampling Strategy

Cosmogenic nuclides in sediments accumulate throughout the sedimentary cycle as near-surface material is exposed during weathering and exposure of the source rock, transport in a specific drainage system, and to a much lesser degree following burial at some intermediate or final destination. Unlike the more commonly used radioactive cosmogenic nuclides, which may decay substantially or even completely over multiple sedimentary cycles, $^{21}$Ne is stable. This means that the concentration of $^{21}$Ne measured in sediments may have accumulated over several cycles of exposure and deposition. For example, after sediments reach the depositional basin, they can be re-exhumed and once again exposed and transported in a new sedimentary cycle. Therefore, the concentration of cosmogenic $^{21}$Ne measured in sediment represents the total exposure during previous and current sedimentary cycles, unless the sediment is exposed during transport to temperatures exceeding the geological closure temperature of Ne in quartz (90-100°C; Shuster and Farley, 2005). The loss of Ne due to diffusion could occur either during burial at depths of ~2-3 km given a geothermal gradient of 30-50°C/km or if rock temperatures reach high enough temperatures for an extended time, which has been recorded in hot desert environments (e.g., McFadden et al., 2005).

We collected and analyzed ten samples in total, eight Hazeva formation samples, and two *in situ* Jordanian cherts. The Hazeva samples include three samples of quartz sand (MHS1, MHS3, and MHS5), and five individual chert pebbles (MHC2, MHC23, MHC5a MHC2b, and MHC6) were obtained from two Miocene Hazeva exposures (Fig. 2 B-C; Table 1). At both sites, samples were collected from deeply shielded locations to minimize the effects of post-burial production (see section 5.1 for further discussion). The quartz sand and the chert pebbles were both transported by the Miocene Hazeva system and share a similar exposure history. However, the quartz sand was exposed in previous sedimentary cycles throughout the Mesozoic and Paleozoic, where it accumulated cosmogenic $^{21}Ne$. In contrast, the chert was deposited in the Eocene and then exposed, transported, and buried during the Miocene (Avni et al., 2012). Therefore, while the cosmogenic $^{21}Ne$ measured in the quartz sand represents multiple sedimentary cycles, the cosmogenic $^{21}Ne$ measured in the chert pebbles represents erosion and transport during a single sedimentary cycle in the Miocene Hazeva River. Additionally, two individual samples of *in situ* chert nodules (EJC3 and EJC5) were collected from exposed bedrock outcrops of the Eocene source rock in central Jordan (Fig. 2A). Unlike the Miocene samples, which were exposed during at least one full sedimentary cycle, the Jordanian chert nodules accumulated cosmogenic nuclides only during exhumation to the currently exposed surface. Therefore, the cosmogenic nuclide concentrations measured in the Jordanian cherts represent averaged rates of erosion over the last $\sim 10^5$ yr.

## 3.2 Preparation of Chert and Quartz Samples and Analytical Procedures

Chert pebbles (ranging 4-14 cm, b axis) were crushed, and both chert and sand samples were sieved to 250-850 μm. Chert and quartz samples were processed to separate clean $SiO_2$ at the Institute of Earth Sciences Cosmogenic Isotope Laboratory, Hebrew University of Jerusalem, following standard procedures (Hetzel et al., 2002; Kohl and Nishiizumi, 1992). The samples were first leached in HCl/HNO$_3$ mixture (3:1) at a temperature of 150°C for 1.5h dissolving carbonates and iron oxides. This procedure was followed by Franz magnetic separation to remove magnetic grains, including quartz grains that contain inclusions of magnetic material. Samples were then leached three times in a 1% HF/HNO$_3$ mixture for 7, 12 and 24h at 70°C, removing the outer rims of the quartz grains. Aliquots of all ten etched samples were then analyzed for Ne isotopes at the Berkeley Geochronology Center. Chert samples were washed with isopropanol to remove fine chert particles attached to the chert grains. Aliquots from samples MCH5A and EJC5 were crushed to compare

the degassing results with the uncrushed aliquots. Ca. 70 mg from the chert samples and ca. 150 mg from the quartz samples were encapsulated in a tantalum packet and heated under vacuum using a diode laser micro-furnace at 2-4 heating steps between 450 and 1250°C for 15 minutes at each temperature step. Ne isotope measurements used the BGC "Ohio" system and the procedure described in Balco et al., (2019). 20-30 grams of leached and clean quartz from three quartz samples and three chert samples were processed to separate Be and Al oxides following Kohl and Nishiizumi (1992) and Bierman and Caffee (2001). These were then analyzed for $^{10}Be/^9Be$ and $^{26}Al/^{27}Al$ at the Centre for Accelerator Mass Spectrometry, Lawrence Livermore National Laboratory, and calibrated against house standards and blanks.

### 3.3 Cosmogenic Scaling and Correction Factors

Exposure and burial times and erosion rates were calculated based on Balco (2007) and scaled using time-independent scaling (Stone, 2000) and production mechanisms based on Balco et al. (2008), given sea-level high-latitude production rates of 4.96 atoms/g $SiO_2$/year for $^{10}Be$, 30.6 atoms/g $SiO_2$/year for $^{26}Al$ (Balco et al., 2008), and 18.1 atoms/g $SiO_2$ year (Borchers et al., 2016; Luna et al., 2018).

### 4. Results

### 4.1 $^{21}Ne$ in Quartz Sand and Cherts

For the chert samples, <2% of the total $^{21}Ne$ and no more than 1% of the total $^{20}Ne$ measured were released above 950°C (see the Supplementary Tables S1-4). Therefore subsequent analyses were performed at 450, 700, and 950°C heating steps for chert samples and 950 and 1250°C heating steps for quartz samples (Table 1). Of the total $^{21}Ne$ measured, >85% was released at the low-temperature steps, below the 950°C step in the chert samples, and below the 1250°C step in the quartz samples (see Supplementary Tables S1-4). Also, low-temperature $^{21}Ne/^{20}Ne$ and $^{22}Ne/^{20}Ne$ ratios fall on the spallation line, within analytical uncertainty. Therefore, we conclude that excess $^{21}Ne$ relative to an atmospheric isotopic $^{21}Ne/^{20}Ne$ ratio of 0.002959 ($^{21}Ne_{ex} = {}^{21}Ne/^{20}Ne_{measured}$- $^{21}Ne/^{20}Ne_{air}$) in the low-temperature steps is a good representation for cosmogenic $^{21}Ne$ ($^{21}Ne_{cos}$; see Supplementary Fig. S8-12). While most samples show some increase in the low-temperature $^{21}Ne_{ex}$, sample MHC2 shows no enrichment in $^{21}Ne/^{20}Ne$ ratio and very little enrichment in $^{22}Ne/^{20}Ne$ ratio compared to atmospheric composition in the low-temperature steps. In the 950°C step, there is enrichment compared to atmospheric values. However, as only ~12% of the total

[21]Ne was released in the 950°C step, determining the concentration of cosmogenic [21]Ne in sample MHC2 is beyond analytical abilities. Therefore, this sample was not considered in further calculations, discussion, and interpretations. It is important to note that even with cosmogenic isotopic values of $^{21}Ne/^{20}Ne$ and $^{22}Ne/^{20}Ne$ ratios at the low-temperature steps, distinguishing the cosmogenic component of $^{21}Ne_{ex}$ from the nucleogenic component, produced by the decay of U and Th within the crystal lattice, is not trivial. Nonetheless, as all chert samples (Eocene chert nodules and Miocene chert pebbles) share the same lithology, any differences in the $^{21}Ne_{ex}$ concentrations must be due to the cosmogenic component.

The chert pebbles and quartz sands sampled at both Miocene Hazeva sites show variable concentrations of $^{21}Ne_{cos}$ ranging between $0.00\pm1.88\cdot10^6$ and $8.89\pm1.83\cdot10^6$ atoms/g $SiO_2$ (Fig. 3). At both Miocene Hazeva sites, the cosmogenic $^{21}Ne$ concentrations measured in chert pebbles are similar or lower compared to sand samples. These measured concentrations agree with our understanding that the sand samples contain quartz grains that originated from various sandy units that were deposited throughout the Phanerozoic and could have undergone several sedimentary cycles before they were exhumed and transported by the Miocene fluvial system. The sand samples could also have higher concentrations of nucleogenic $^{21}Ne$ as the source rock for this sand is >800 Ma (Kolodner et al., 2009). Conversely, the Hazeva chert samples are derived from a relatively young Eocene source rock and were exposed during one sedimentary cycle in the Miocene. Both samples of Jordanian chert nodules collected from *in situ* Eocene outcrops show similar cosmogenic **[21]**Ne concentrations, higher compared to the Miocene Hazeva chert pebbles (Fig 3).

Diffusion kinetics of Ne in quartz have been examined experimentally and theoretically (Shuster and Farley, 2005; Tremblay et al., 2014) but have yet to be tested on chert samples, where the diffusion length-scale is not straightforward. While diffusion kinetics in chert are likely to be similar to quartz, more work is needed to determine that with certainty. Nevertheless, diffusion is not likely to have been significant over a ~20 Myr timespan in the measured Miocene chert samples. While temperatures in exposed cherts in the Levant region can reach 60-70°C during mid-day in the summertime due to solar heating, it is unlikely that samples that were transported fluvialy were exposed continuously at the surface. The examined chert samples did not exhibit any visible cracking or fractures commonly identified with thermal stresses, leading us to believe that temperatures were not high enough to cause significant diffusion of Ne out of the chert samples.

## 4.2 $^{10}$Be and $^{26}$Al in Quartz Sand and Cherts

$^{10}$Be and $^{26}$Al concentrations were measured in three Miocene sand samples (MHS1, MHS3, and MHS5), the two Eocene chert nodules (EJC3 and EJC5), and two chert pebbles (MHC5b and MHC6). $^{10}$Be results for sample MHC5b and $^{26}$Al results for sample MHS1 are not available (Table 1). Miocene sand and chert samples show $^{10}$Be and $^{26}$Al concentrations that are low and consistent with extended periods of burial ($\leq 0.39.\pm 0.03 \cdot 10^5$ atoms/g SiO$_2$ for $^{10}$Be and $\leq 4.33.\pm 0.55 \cdot 10^5$ atoms/g SiO$_2$ for $^{26}$Al). Currently eroding Eocene nodules show higher concentrations of $^{10}$Be and $^{26}$Al, with sample EJC3 showing $^{26}$Al/$^{10}$Be ratio that is consistent with production at the surface, and sample EJC5 showing a lower $^{26}$Al/$^{10}$Be ratio, suggesting a more complicated exposure history (see Discussion section).

## 5. Discussion

### 5.1 Correcting for Post-Burial Muonic Produced Cosmogenic $^{21}$Ne

When examining concentrations of cosmogenic nuclides in sediments that have been buried for extended periods, post-burial production needs to be considered. At or near the surface, spallation interactions are the main pathway for *in situ* production of cosmogenic nuclides accounting for >95% for $^{26}$Al, $^{10}$Be, and $^{21}$Ne (Dunai, 2010). However, the relative contribution of production by muon interactions increases with burial depth. While production rates are relatively low, they can be significant when integrated over long periods, especially for stable nuclides. The post-burial component does not represent surface processes, and therefore, it is crucial to account for its contribution to the measured cosmogenic component. For radioactive cosmogenic nuclides, such as $^{10}$Be and $^{26}$Al, their initial concentrations (acquired during exposure) decrease post burial due to radioactive decay, with $^{26}$Al decreasing faster than $^{10}$Be according to their corresponding half-lives (e.g., Balco and Rovey, 2008; Granger, 2006; Granger and Muzikar, 2001; Lal, 1991).

We calculated the expected concentrations of cosmogenic $^{26}$Al, $^{10}$Be, and $^{21}$Ne in sediments over a burial period of 18 Myr, the likely age of the fluvial system stabilization (Bar and Zilberman, 2016). We then compared these calculated concentrations to the measured concentrations of $^{26}$Al, $^{10}$Be, and $^{21}$Ne$_{cos}$ in Miocene chert and sand samples (Fig. 4). Both $^{10}$Be and $^{26}$Al measurements are only available for two buried sand samples, one buried chert pebble, and two *in situ* chert nodules (Table 1). The measured $^{10}$Be and $^{26}$Al concentrations have reached an equilibrium that is consistent with an extended period of burial at depths between 20-120 m (given that overburden

consists of clastic sediments with a density of ~2 g/cm$^3$). The discrepancy between the current

burial depth, only tens of meters below the surface, and the deduced burial depth is likely the result of surface erosion that occurred during the last ~2 Myr (Matmon and Zilberman, 2017 and references therein). Additionally, the relatively large uncertainty on muogenic production rates could account for some of this discrepancy (Balco, 2017; Balco et al., 2019). Our calculations show that the cosmogenic $^{21}$Ne produced post-burial over 18 Myr at depths between 20-120 m is

lower than the $^{21}$Ne$_{ex}$ measured in the presented samples (including their uncertainties). The maximal calculated post-burial cosmogenic $^{21}$Ne concentration accounts for ~$1.3 \cdot 10^6$ atoms/g SiO$_2$, which is lower than the analytical uncertainty for all measured Miocene samples except for MHC2, where no cosmogenic $^{21}$Ne was measured. However, sample MHC2 is not considered in the interpretations of the results. Therefore, we consider post-burial cosmogenic $^{21}$Ne production

to be insignificant for the presented Miocene exposure times.

**5.2 Calculating Modern and Miocene Exposure Times**

Exposure times at the surface calculated from cosmogenic $^{21}$Ne concentrations measured in *in situ* chert nodules from the Jordanian Central Plateau (EJC3 and EJC5) range between a minimum of 193 kyr and a maximum of 454 kyr (correlating to cosmogenic $^{21}$Ne concentrations of

$8.08 \pm 1.48 \cdot 10^6$ and $12.10 \pm 2.43 \cdot 10^6$ atoms/g SiO$_2$).

In comparison to the Jordanian samples, quantifying exposure times during the Miocene using cosmogenic $^{21}$Ne concentrations is not trivial, most notably due to the challenge in evaluating the local cosmogenic production rates. The production rate of cosmogenic nuclides increases with altitude as the air pressure and shielding effect of the atmosphere decreases (Stone, 2000). While

the latitude of the Arabian Peninsula during the early Miocene was similar to today (Meulenkamp and Sissingh, 2003, and references therein), accounting for the elevation of the Miocene samples during the production of cosmogenic $^{21}$Ne raises two difficulties. First, it is not possible to determine with certainty the elevation of the Jordanian Central Plateau during the Miocene. It is clear that from the Late Cretaceous up until the late Eocene, the Arabian Peninsula was mostly

submerged below sea level and that during the Oligocene, it was uplifted to a sufficient elevation to allow for significant surface erosion (Garfunkel, 1988). During the early Miocene, broad valleys (500-1000 m wide and ~100 m deep) incised the regional truncation surface that developed in the region, where the Hazeva formation was later deposited (Avni et al., 2012). This timeline of events leads us to believe that significant surface uplift occurred prior to the initiation of the Miocene

Hazeva fluvial system at ~18 Ma. Nevertheless, this stratigraphic evidence is insufficient to determine whether the Arabian Peninsula reached its current elevation during the early-mid Miocene or whether additional uplift occurred over the past 20 Myr. Studies that focus on exhumation along the eastern flank of the Dead Sea Rift do not provide clear evidence to constrain the timing of surface uplift. Surface uplift histories based on cooling ages (Feinstein et al., 2013), and river profiles (Wilson et al., 2014), conclude that during the last ~30 Myr the western half of the Arabian Peninsula was uplifted to its current elevation(Feinstein et al., 2013; Wilson et al., 2014). However, in a recent work, Morag et al. (2019) present thermochronologic constraints using apatite (U–Th)/He and fission-track data from a transect across the eastern flank of the Suez Rift in SW Sinai. The researchers suggest that uplift and exhumation along the western side of the Suez Rift flank slowed substantially post ~18 Ma. This decline reflects a decrease in uplift, which could indicate that the Jordanian Plateau has reached close to its current elevation (~1000 m) when the Hazeva River was active. One more approach to evaluate the paleo-elevation of the Central Jordanian Plateau is to calculate this elevation given a known distance between the source point and the base level, and an evaluated slope. The Hazeva fluvial system drained westward to the Mediterranean at an elevation of ~0 m.a.s.l, and over a distance of ~200 km from the Mediterranean coast to the location of exposed chert nodules. Given a moderate stream gradient of ~0.5%, the elevation of the Central Jordanian Plateau is ~1 km above sea level. Given the different types of evidence reported, it is reasonable to presume that the western flank of the Arabian Peninsula reached its current elevation (~1 km) during the early-mid Miocene. The use of a single elevation to calculate paleo-production rates introduces a second difficulty, as it does not account for spatial variations in elevation due to catchment topography. Without any tangible information about the size and steepness of the catchment area of the Hazeva River, we are unable to correct for different elevations and production rates throughout the basin. These uncertainties in paleo-production rates, due to assumptions in catchment paleo-elevation, result in longer calculated exposure times. Accounting for uncertainties described above, we assume an elevation range of 500-1000 meters above sea level, and latitude of 20-30° for the calculated Miocene exposure times.

The calculated exposure times of sediments in the Miocene Hazeva fluvial system are variable and range between a minimum of $0_{-0}^{+59} - 0_{-0}^{+86}$ kyr measured in chert pebble sample MHC5b and a maximum of 278±63 – 408±63 kyr measured in quartz sand sample MHS5 (Table 2). Comparing

the two silicate members, concentrations (and exposure times) of the sand samples are overlapping or higher than the chert samples (Fig. 3). This observation agrees with our understanding that the cosmogenic $^{21}$Ne measured in the Miocene chert pebbles represents the total time of exposure during exhumation from bedrock coupled with transport in the Hazeva River. At the same time, the sand samples have undergone previous sedimentary cycles and contain inherited cosmogenic $^{21}$Ne. Therefore, sand samples cannot be used to calculate the time sediments were exposed during transport in the Hazeva fluvial system or to infer erosion rates. Unlike the sand samples, that have feasibly undergone previous exhumation, erosion, and deposition, the Miocene chert samples have not undergone previous sedimentary cycles. Hence, all cosmogenic $^{21}$Ne measured was produced during erosion and transport in the Hazeva River and rates of surface processes during the Miocene can be evaluated using the Miocene chert samples.

The cosmogenic $^{21}$Ne exposure times calculated from the Jordanian chert samples range from 269±63 to 378±76 kyr. Exposure times that were calculated from $^{10}$Be and $^{26}$Al concentration measured in sample EJC5 overlap within uncertainty with $^{21}$Ne calculated exposure values (Table 2). In contrast, exposure times calculated from $^{10}$Be and $^{26}$Al concentrations measured in sample EJC3 are much shorter ~13-16 kyr, an order of magnitude difference. While we cannot explain this discrepancy, we believe that the representative results are longer exposure times. Firstly, the $^{21}$Ne calculated exposure time in sample EJC3 agrees with the $^{21}$Ne, $^{26}$Al, and $^{10}$Be calculated exposure times for sample EJC5. Secondly, the timescales of exposure times measured in cherts in eroding surfaces at hyperarid Negev Desert are similar and range from ~$2 \cdot 10^5$ to ~$2 \cdot 10^6$ yr (Boroda et al., 2014; Fruchter et al., 2011; Matmon et al., 2009). We conclude that exposure times in modern Jordanian Central Plateau chert nodules range ~300-400 kyr. It is important to note that the calculated exposure times in the Jordanian cherts represent only exposure at the surface, and do not include exposure during transport, in contrast to the Miocene chert pebbles.

When examining ancient exposure times, we must first consider the time-scales over which cosmogenic nuclides are averaged. The question arises whether the reported exposure times accurately represent the environmental conditions of a certain period (e.g., the early to mid-Miocene) or if the calculated times are the result of episodic oscillation or catastrophic geomorphic events. For currently exposed *in situ* samples, the modern exposure times are relatively long, integrating hundreds of thousands of years, over which such oscillations or rare catastrophic events would be averaged. As for the Miocene exposure times, samples were collected from two separate

sites and different depths, so it is unlikely that they all represent the exception. We, therefore, consider the range of times obtained from Miocene samples to be a good representation of Miocene surface processes.

**5.3 Modern and Miocene Erosion Rates and the Influence of Climate and Tectonics**

The calculated exposure times of the Jordanian chert nodules are equivalent to erosion rates of ~4-12 mm/kyr (Table 2), consistent with other rates measured in the region (Matmon and Zilberman, 2017 and references therein). Calculation of paleo-erosion rates is not as straightforward, as Miocene cherts were sampled post-deposition and represent exposure both during erosion from bedrock and transport in the Hazeva River. However, Miocene exposure times are either shorter or overlap within uncertainty with times of *in situ* Jordanian chert. Thus, actual bedrock erosion rates during the Miocene must have been faster than the prevailing rates mentioned above.

While we cannot determine how much faster paleo-erosion rates were during the Miocene, any increase in erosion rates in a hyperarid desert must be the consequence of different environmental conditions that prevailed in the region at that time. An increase in rates of erosion is most commonly attributed to perturbations in fluvial basins in response to tectonic uplift and/or warmer/wetter climatic conditions (e.g., DiBiase and Whipple, 2011; Romans et al., 2016; Schaller and Ehlers, 2006; Val et al., 2016; Willenbring et al., 2013). For example, increased precipitation brings about higher river discharge and enhancement of the stream power available for bedrock erosion and sediment transport. Erosion rates in fluvial systems also respond to tectonically induced changes in base level that increase slope steepness and instability, resulting in higher stream power and more sediment readily available for transport. Here we examine evidence from previous studies of the climatic and tectonic conditions that prevailed in the region during the Miocene, capable of forcing the deduced increase in erosion rates.

Many works which quantify the rates and timing of surface uplift related to the rifting of the Red Sea are confined to the edges of the Arabian plate and do not give good constrains for intercontinental uplift (Morag et al., 2019; Omar et al., 1989; Omar and Steckler, 1995). These studies used thermochronometric methods and focused on the uplifted flanks of the Suez Rift along which the Precambrian basement of the Arabian-Nubian Shield is exposed. Constraining uplift of the Arabian Plateau is more challenging as the exposed strata are composed mostly of carbonate rocks, which are not suitable for this type of methods. While some studies point to a decrease in

exhumation rates during the mid-Miocene (~18 Myr; Morag et al., 2019), surface uplift and topographic changes could still drive large-scale landscape response, manifesting as increased erosion rates and the establishment of the Hazeva fluvial system.

In addition to tectonic forcing, there is ample evidence for a warmer and wetter climate in the region during the Miocene. Locally, the appearance of mammals in the Negev, along with arboreal and grassy vegetation during the early-mid Miocene, supports a humid environment (Goldsmith et al., 1988; Horowitz, 2002; Tchernov et al., 1987). Tropical to subtropical climate prevailed in the eastern Arabian Peninsula, as indicated by fossilized mangrove roots (Whybrow and McClure,

1980). Locally, Kolodny et al. (2009), interpreted the $^{18}O$ in lacustrine limestone from the lower part of the Hazeva unit to be deposited by $^{18}O$-depleted paleo-meteoric water. They proposed that the presence of a warm ocean to the southeast of the region during the Late Oligocene-Early Miocene resulted in tropical cyclones being more prevalent and increasing rainfall in the region. Together, the above observations suggest climatic conditions, which could promote erosion rates

that are faster than observed rates in hyperarid conditions, and that support the existence of a large and maintained fluvial system, such as the Hazeva River, during the Miocene.

## 6. Conclusions

We compared the cosmogenic $^{21}Ne$ measured in chert pebbles and quartz sand eroded and transported during the mid-Miocene (~18 Myr) by the Hazeva River with the chert source rock

(Eocene chert nodules) currently eroding in the Central Jordanian Plateau.

We successfully established a novel application for measuring cosmogenic $^{21}Ne$ in modern and Miocene chert samples, expanding the opportunities and settings in which stable cosmogenic nuclides analysis could be used as a tool to quantify geomorphic processes and ascertaining chert as a viable lithologic target for cosmogenic Ne analysis. In modern samples, measurements of

cosmogenic nuclides $^{10}Be$ and $^{26}Al$ generally agree with $^{21}Ne$ results. In the Miocene samples, cosmogenic $^{21}Ne$ in quartz sand samples is equal or higher compared to Miocene chert pebbles, agreeing with the geologic understanding that sand has experienced several sedimentary cycles where $^{21}Ne$ was produced. In contrast, chert experienced only one such cycle in the Miocene Hazeva fluvial system.

Exposure times calculated from the measured cosmogenic $^{21}Ne$ concentrations in the Miocene chert pebbles are shorter compared to the chert nodules currently eroding in the Central Jordanian

Plateau. While it is impossible to determine the exact rate of erosion during the Miocene, as cosmogenic $^{21}$Ne was produced during erosion from the bedrock and transport in the river, shorter exposure times during the Miocene point to rates of surface erosion being faster. The cause for

increased rates during the early-mid Miocene cannot be easily constrained to either tectonic or climatic conditions. The entire region experienced tectonic uplift and exhumation that, while possibly decreasing during the mid-Miocene, brought on topographic changes that established the Hazeva fluvial system and could have manifested as faster rates of surface erosion. Furthermore, multiple independent proxies presented in previous studies support wetter climatic conditions in

the region during the early-mid Miocene. Increased precipitation would explain the faster rates of bedrock erosion deduced as well as the higher water discharge needed to maintain transport along the Hazeva River. Finally, the variability observed in exposure times of Miocene chert pebbles might represent a change in rates of erosion throughout the Miocene. However, this variability in $^{21}$Ne concentrations is more likely the result of fluvial transport dynamics, temporary storage, and

exposure during transport in this large Miocene river.

**Data availability**

A raw data table, including all Ne isotope measurements, and three-isotope plots are available in the supplement.

**Author contribution**

MBI and AM designed the study. MBI collected the samples for analysis with assistance from AM and YA. MBI prepared samples for analyses and measured $^{21}$Ne/$^{20}$Ne and $^{22}$Ne/$^{20}$Ne ratios with GB, and AJH measured the $^{10}$Be/$^{9}$Be and $^{26}$Al/$^{27}$Al ratios. MBI analyzed the data, produced the figures, and prepared the manuscript with contributions from all co-authors.

**Competing interests**

The authors declare that they have no conflict of interest.

**Acknowledgments**

This work was funded by the Israel Science Foundation (*ISF* grant number 385/14 to AM) and further supported by the United States-Israel Binational Science Foundation (BSF travel grant T-2017229 to MBI). We greatly appreciate the intensive work and insightful comments by Prof.

Taylor Schildgen, Dr. Marissa Tremblay, and an anonymous reviewer. Our gratitude to Y. Geller, O. Tirosh, and Y. Burstyn for laboratory and field assistance. MBI would like to thank the technical and administrative staff at the Berkeley Geochronology Center for their assistance and support. This work was performed in part under the auspices of the U.S. Department of Energy by Lawrence Livermore National Laboratory, United States under Contract DE-AC52-07NA27344. This is

LLNL-JRNL-788357.

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

**Table 1: Sample Description, Sampling Site Locations and Cosmogenic Nuclide Data**

| Sample | Sample type | Site | Sampling depth below surface (m) | Location Lat (°N) | Long (°E) | Elevation (m.a.s.l) | Be Carrier (mg) | $^{10}Be/^9Be$ ($\times10^{-13}$) | $[^{10}Be]$ ($10^5$ atoms/ g SiO$_2$) | $^{26}Al/^{27}Al$ | [Al]* (ppm) | $[^{26}Al]$ $10^5$ atoms/ g SiO$_2$) | Al/Be | $[^{21}Ne_{cos}]$† |
|---|---|---|---|---|---|---|---|---|---|---|---|---|---|---|
| MHS1 | Quartz sand | Paran Valley, Israel | 30 | 30.33296 | 34.92724 | 290 | 176 | 0.17±0.03 | 0.14±0.02 | NA | 104 | NA | NA | MHS1 |
| MHS3 | Quartz sand | Arad Quarry, Israel | 90 | 31.23372 | 35.20685 | 570 | 171 | 0.36±0.02 | 0.29±0.02 | 0.60±0.08 | 110 | 1.33±0.17 | 4.57±064 | MHS3 |
| MHS5 | Quartz sand | Arad Quarry, Israel | 100 | 31.23372 | 35.20685 | 570 | 175 | 0.32±0.02 | 0.26±0.02 | 0.35±0.04 | 114 | 0.86±0.11 | 3.25±0.44 | MHS5 |
| MHC2 | Chert pebble | Paran Valley, Israel | 20 | 30.33296 | 34.92724 | 290 | NA | NA | NA | NA | NA | NA | NA | MHC2 |
| MHC3 | Chert pebble | Arad Quarry, Israel | 90 | 31.23372 | 35.20685 | 570 | NA | NA | NA | NA | NA | NA | NA | MHC3 |
| MHC5a | Chert pebble | Arad Quarry, Israel | 100 | 31.23372 | 35.20685 | 570 | NA | NA | NA | NA | NA | NA | NA | MHC5a |
| MHC5b | Chert pebble | Arad Quarry, Israel | 100 | 31.23372 | 35.20685 | 570 | 172 | NA | NA | 0.93±0.12 | 203 | 4.33±0.55 | NA | MHC5b |
| MHC6 | Chert pebble | Paran Valley, Israel | 30 | 30.33296 | 34.92724 | 290 | 170 | 0.10±0.01 | 0.39±0.03 | 0.05±0.02 | 287 | 0.32±0.13 | 0.83±0.35 | MHC6 |
| EJC3 | In situ chert | Central Jordanian Plateau | Surface | 30.97045 | 36.64469 | 910 | 172 | 0.70±0.03 | 1.13±0.05 | 1.50±0.10 | 230 | 6.81±0.43 | 5.11±0.38 | EJC3 |
| EJC5 | In situ chert | Central Jordanian Plateau | Surface | 30.87181 | 36.52129 | 1000 | 178 | 18.43±0.30 | 29.75±0.49 | 11.47±0.25 | 235 | 72.96±1.54 | 2.45±0.07 | EJC5 |

*Note:* NA – not available. Samples were either not analyzed, or no result was attained.

*Measurement uncertainties are ~5%.

†Cosmogenic $^{21}Ne$ is the excess of $^{21}Ne$ concentrations relative to the atmospheric $^{21}Ne/^{20}Ne$ ratio, calculated for the low-temperature steps (<950°C for chert and <1250°C for quartz).

**Table 2: Exposure times and erosion rates calculated for the modern and Miocene samples**

| Sample | Sample type | Location | Exposure time (kyr) | Erosion rate (mm/kyr) |
|---|---|---|---|---|
| MHS1 | Miocene quartz sand | Paran Valley, Southern Negev Desert | 114±46 – 166±87 | - |
| MHS3 | Miocene quartz sand | Arad Quarry, Northeastern Negev Desert | 280±10 – 408±63 | - |
| MHS5 | Miocene quartz sand | Arad Quarry, Northeastern Negev Desert | 278±17 – 404±83 | - |
| MHC3 | Miocene chert pebble | Arad Quarry, Northeastern Negev Desert | 167±53 – 242±113 | 3.0±1.4 – 4.4±1.4 |
| MHC5a | Miocene chert pebble | Arad Quarry, Northeastern Negev Desert | 91±46 – 132±78 | 5.5±3.3 – 8.0±4.7 |
| MHC5b | Miocene chert pebble | Arad Quarry, Northeastern Negev Desert | $0^{+59}_{-0} - 0^{+85}_{-0}$ | >8.6 – >12.4 |
| MHC6 | Miocene chert pebble | Paran Valley, Southern Negev Desert | 121±59 – 176±102 | 3.0±1.4 – 4.4±3.5 |
| EJC3[*] | In situ chert nodule | Central Jordanian Plateau | 269±49 / 16±1 / 13±1 | 2.7±0.5 / 41.7±1.7 / 50.0±3.2 |
| EJC5[*] | In situ chert nodule | Central Jordanian Plateau | 378±76 / 361±6 / 378±3 | 1.9±0.4 / 1.7±0.0 / 4.4±0.1 |

*Note:* Exposure times is the 'simple exposure time' calculated for exposure at the surface, calculated cosmogenic $^{21}$Ne production rates ranging 22.2-30 (atoms/g $SiO_2$ yr), given an elevation of 500 and 1000 meters above sea level. Erosion rates for sand samples were not calculated as the concentration of cosmogenic $^{21}$Ne might include inherited cosmogenic $^{21}$Ne from previous sedimentary cycles.
[*]Erosion rates calculated using $^{21}$Ne / $^{10}$Be / $^{26}$Al.

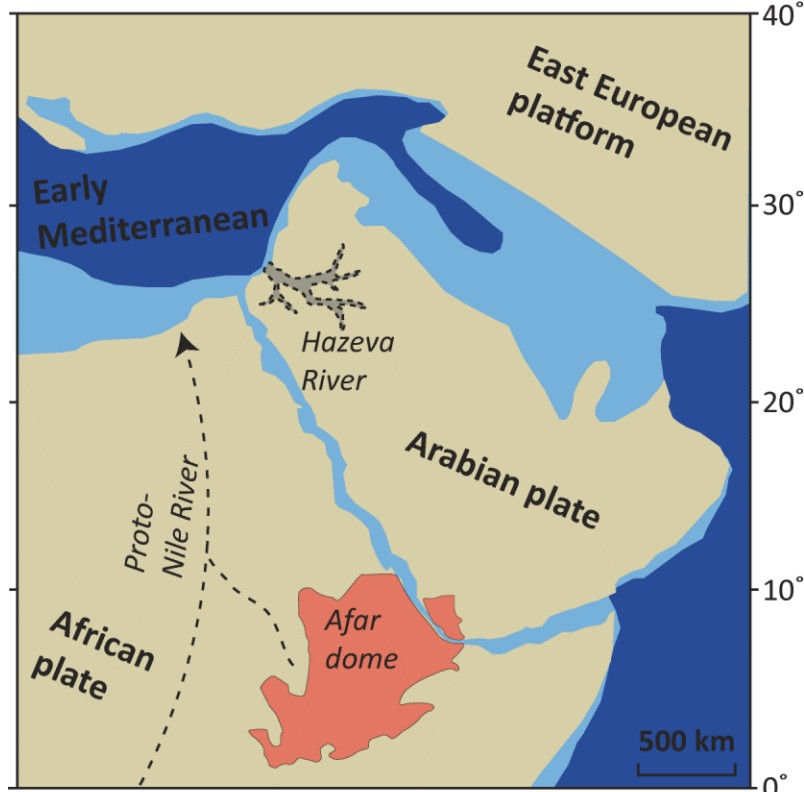

**Figure 1. Paleo-geographic map of the eastern Levant during the early Miocene (modified after Meulenkamp and Sissingh, 2003) with the approximated extent of the Hazeva fluvial system (based on Avni et al., 2012; Zilberman and Calvo, 2013).**

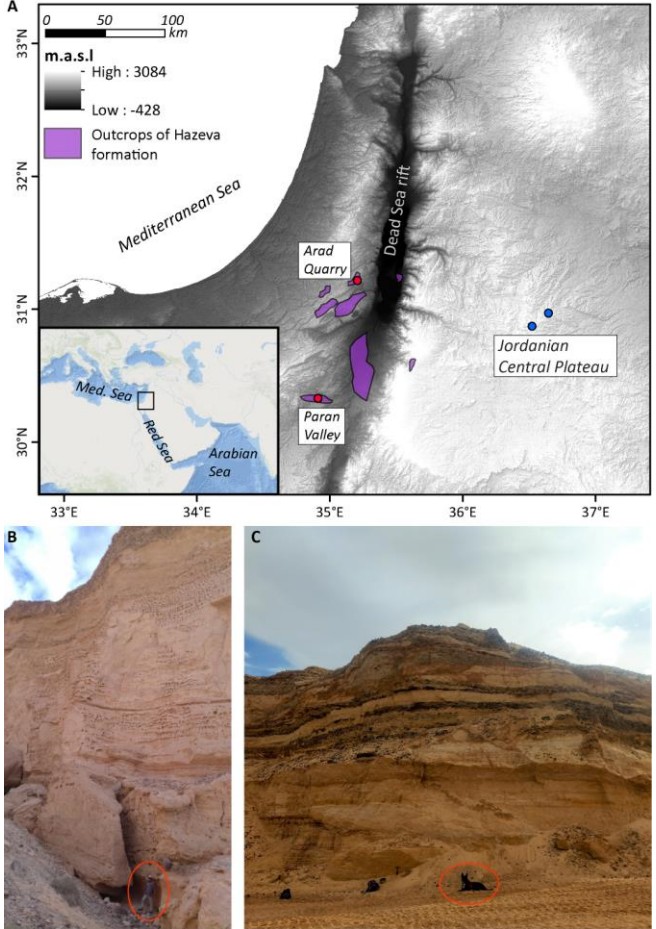


**Figure 2. (A)** Shaded relief map of the study area with sampling locations of Miocene fluvial sediments (red) and in situ Eocene source rock (blue). Hazeva outcrops are after Zilberman and Calvo (2013). The inset map shows the regional geographical context. **(B)** Sampling location at Paran Valley. Sample collected from behind the fallen boulder in a narrow canyon and underneath an
overburden of ~50 meters of sand and conglomerate. See person for scale marked at the bottom. **(C)** Photo of sampling location at Arad Quarry. Samples collected from underneath an overburden of ~100 meters of quartz sand. See dog for scale marked at the bottom.

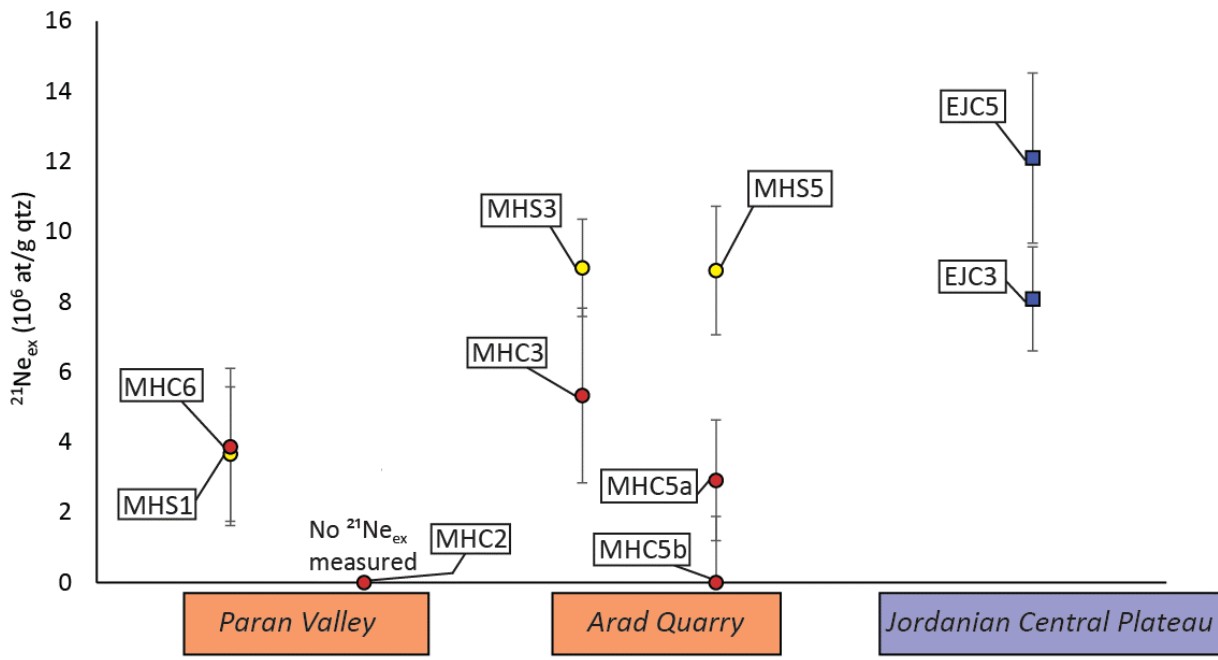

**Figure 3. $^{21}Ne_{cos}$ concentrations in Hazeva sands (yellow), Hazeva chert pebbles (red), and *in situ* Jordanian Central Plateau chert nodules (blue) with respective uncertainties.**

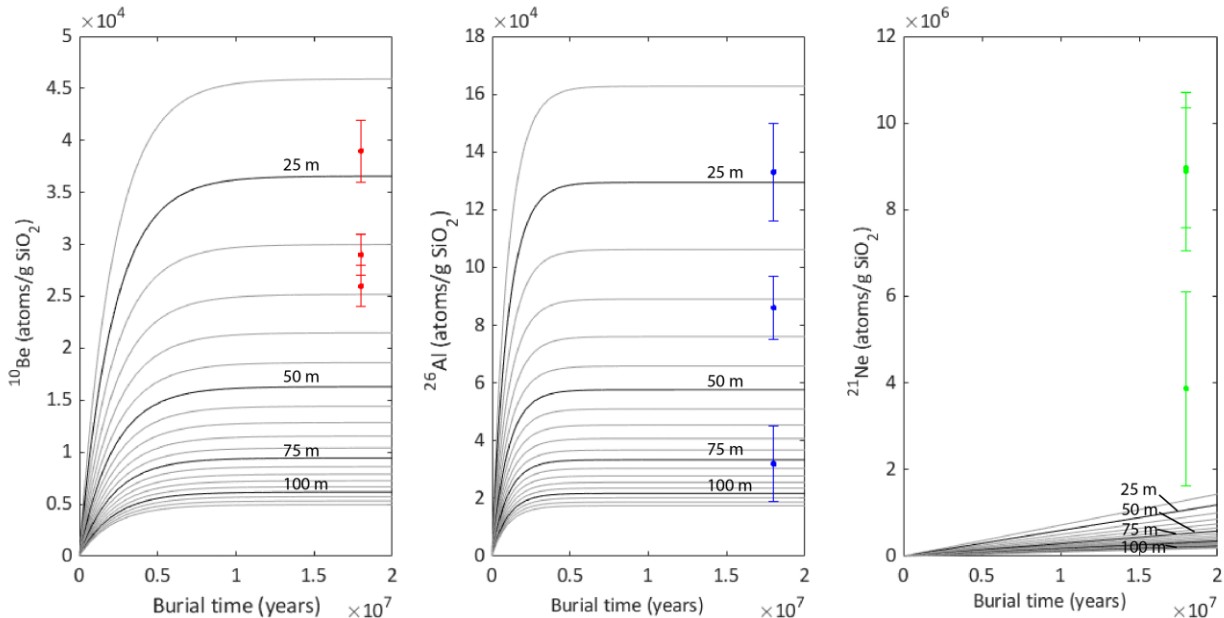

**Figure 4. Measured concentrations of $^{10}$Be (red), $^{26}$Al (blue), and $^{21}$Ne (green) in samples MHS3, MHS5, and MHC6. Grey contour lines show changes in nuclide concentrations with time at different depths from 20 to 120 m below the surface in 5m increments. For both sand samples and the chert sample, the concentrations of cosmogenic $^{21}$Ne are higher than the estimated post burial production. Production by cosmic-ray muons is calculated with schematics presented by Balco (2007). Production rates were calculated at the Arad Quarry site by cosmic-ray muons of $^{10}$Be and $^{26}$Al are after Balco (2017) and of $^{21}$Ne by fast muons is after Balco et al. (2019). This illustration shows that $^{10}$Be and $^{26}$Al concentrations can be explained by post-burial production, but $^{21}$Ne concentrations cannot, so a significant fraction of cosmogenic $^{21}$Ne is pre-burial.**

