# Peer review of "Early-mid Miocene erosion rates inferred from pre-Dead Sea rift"

_Earth Surface Dynamics, 2019_

## Referee Comment (RC1) · Taylor Schildgen (Referee) · 7 Oct 2019

Ben-Israel and co-authors analyze stable (21Ne) and unstable (10Be, 26Al) cosmo-genic nuclides in detrital Miocene sediments from the NW Arabian plateau to calculate paleo-erosion rates and compare these to "modern" rates obtained from bedrock out-crops. They interpret an approximate 2-fold difference in concentrations to result from erosion rates that were 2x faster during the early-mid Miocene, which could be reason-able considering evidence for a wetter climate in the region at that time. On the positive side, this work illustrates the unique ability of the stable cosmogenic nuclide 21Ne to record erosion rates averaged over relatively short time intervals (if we can consider

100s of kyr short) in ancient, well-shielded sedimentary deposits; such information cannot be obtained with 10Be or 26Al. The authors are careful in their consideration of various potential complications of their data – post-burial deposition, changes in elevation through time, and how different types of detrital material (quartz sand v. chert pebbles) could have experienced very different pathways to the final deposition site. I also really like the use of different types of detrital material to assess the possibility of 21Ne inheritance. But I see several areas that require improvement. Most concerning for me is the discussion of the uplift-history constraints, which I don't find very convincing, but are critical for the final interpretations, as the difference in measured 21Ne concentrations between modern and mid-Miocene samples can be explained either by a change in elevation through time or a change in erosion rates. But even if there has not been a change in elevation, the difference in erosion rates reported (2-4 mm/kyr v. 4-12 mm/kyr) is not huge, and those mid-Miocene rates are still pretty darn slow. Is this really a story about how climate affects erosion rates, or could the conclusion be that it doesn't affect them all that much?

I'm also concerned by the small number of samples obtained from the outcrops (only two), and the possibility that the rates reported are not representative of modern rates (often spot samples from outcrops lead to a wide range of erosion-rate estimates). Are there any other modern erosion rates that have been reported that can be used to corroborate the results presented here? Is there a reason a modern erosion-rate estimate wasn't made from modern detrital sands in the region, even if that rate would not be from exactly the same drainage area as the early-mid Miocene samples?

I think the difference between the detrital quartz sand and detrital chert pebbles can be better emphasized in the final interpretations/conclusions of the paper. It seems that the quartz-sand results are not considered in the final interpretations due to the possibility that the quartz experienced multiple periods of deposition and exposure prior to the last deposition, hence it contains inherited 21Ne (if that's the reason that the erosion rates for the quartz samples are not reported, the authors should state

that explicitly rather than leaving it for the readers to infer; still I think the erosion rates should be reported). But rather than making it seem as if those samples were just a waste of time, it could be helpful to emphasize how in recycled sediments, inherited 21Ne can be a real problem. I like the approach here of measuring different types of detrital material to assess this possibility! That could be highlighted, rather than hidden.

Finally, I think the introductory paragraph can be improved; several of my line-specific comments refer to my confusion about where the paper is going just in the first several sentences.

Line-specific comments:

l. 36-37: Older landscapes are transient? Odd wording. Also, this sentence doesn't really follow from the previous ones. You've discussed river systems and sediment archives, now we're on to preservation of landscapes themselves? Be more precise and focused.

l. 38: Okay, so the focus is on quantifying erosion rates from surfaces? This is not easy to follow.

l. 41: If the focus is on erosion, don't change the terminology here to "surface processes", as that encompasses much more than just erosion.

l. 43: Now you've explained that the focus is on sedimentary deposits, not slowly eroding surfaces. I suggest rewriting this whole paragraph with a clearer focus on what information you want to give to the reader. What is the main problem, why is it difficult to address, how are you going to do it?

l. 51: Wouldn't it be the other way around, i.e., the Afar plume leads to magmatic events, and maybe even influenced tectonics?

l. 67-69: This means that the deposition associated with the river started prior to ca. 20 Ma? Or do you interpret only the upper part of the Hazeva Formation to be associated with the river, meaning that fluvial deposition started after 20 Ma? Please clarify.

l. 99-101: How is this history of the quartz sand known? If this history is going to be important for explaining differences between 21Ne measured in quartz vs. chert, then a fuller explanation is needed. One potential worry with quartz sand is that it could be aeolian in origin; can this be ruled out? What size fraction of sand was processed?

l. 90-105: I suggest moving this paragraph to the geological setting, as it provides the geological context for the samples collected.

l. 109: How deeply shielded were the collected samples? Deeply enough to rule out post-depositional 21Ne production?

l. 113: I suggest "accumulated cosmogenic nuclides only during exhumation", as the samples that experienced the full sedimentary cycle also accumulated nuclides during exhumation.

l. 156-157: Is this because you assume the U and Th are equally distributed throughout the rock? Is that a reasonable assumption?

l. 212-217: Don't assume that your readers remember that EJC5 and EJC3 were the surface samples of from the "in situ" outcrops, remind us.

l. 214-215: This detail concerning the scaling of production rates belongs in the methods, not the discussion.

l. 223-224: I don't see the added value of reporting equivalent exposure times (if that is what is meant by "simple exposure time"), given that you are mainly interpreting the measured concentrations in terms of erosion rates. Or is the goal to give readers a sense of the averaging timescale of these erosion rates? If the latter, I suggest rewording to make this clear.

l. 224-226: It is important, but why? I can make a guess, but it would better if you explain.

l. 230-231: Could you please briefly remind me what those differences in concentrations are?

l. 237-239: A bigger overview map that includes the Suez rift in addition to the all the other relevant sites mentioned here would be very helpful.

l. 237-242: These uplift constraints are crucial for your interpretation of whether or not there the Miocene samples show a faster erosion rate compared to today or reflect a similar erosion rate with a lower nuclide paleo-production rate. Given their importance, some more details on these uplift constraints would be very helpful. Although I have not checked each of the references in detail, I disagree with how you have referenced the Wilson et al. (2014) interpretations. Despite many reasons why these interpretations of uplift histories from river profiles should be considered suspect, their interpretation for your field area is that most of the modern elevation gain occurred since 20 Ma, and it looks like more than half of that is since 10 Ma (see their Fig. 17). For that reason, I don't agree at all with your statement that it is reasonable to presume that the western flank of the Arabian Peninsula (or the NW edge, corresponding to your field area) reached its current elevation prior to the initiation of the Hazeva fluvial system at ca. 18 Ma.

l. 255-258: I can guess why you do not mention erosion rates from the quartz sand – because you suspect it has inherited 21Ne – but it seems like an oversight. I suggest to not "hide" those results, but rather emphasize how recycling of quartz sand can lead to incorrect results.

l. 281-287: Mostly I've been able to work out myself whether you are referring to surface uplift or rock uplift up until now, but in this section in particular I cannot follow your meaning. In this paragraph, and ideally throughout the manuscript, specify which one you are referring to.

l. 285: But this is not an accurate representation of the uplift history for your field area as interpreted by Wilson et al. (2014).

l. 288-300: This evidence for a wetter early to mid Miocene climate seems reasonable, and I agree that such a climate would likely erode faster than low-relief, hyperarid landscapes. But the mid-Miocene erosion rates reported here, which might be considered maximum rates given the uncertainty in the paleo-elevation, still seem very slow. How do rates of 4 to 12 mm/kyr compare with erosion rates measured from similar environments today? (Incidentally, I realize I'm assuming that the landscape relief is relatively low, but it would be helpful to actually show a slope/relief map to see whether or not that's the case.)

Figure 2: As mentioned above, a broader overview map would be very helpful. In 2B, is that a person near the bottom? Highlighting or circling him/her in some way would make it easier to understand the scale of this photo. Likewise, in 2C, is that a dog?

Figure 3: Given the overall focus on erosion rates, I find it odd that the calculated erosion rates are not shown in this figure. Why not use those instead of the effective exposure ages?

Editorial comments:

l. 35: always specify what you mean after "this", e.g., this lack of information?

l. 64: "comprise" rather than "compose"

l. 71: I'd suggest "disruption" rather than "dismantlement"

l. 81: What "cycles" are you referring to here? Okay, you clarify it in the next sentence, but please instead clarify at your first instance of using this term.

l. 86: This what should hold true?

l. 96: comprising, not composing

l. 210: Please refer to "denudation" or "erosion" rates throughout, not "rates of surface processes", which is unnecessarily vague.

l. 221: lots of needless words here, please shorten to "erosion rates between 1 and 5 mm/kyr"

---

## Referee Comment (RC2) · Anonymous Referee #2 · 30 Oct 2019

Review Ben-Israel et al.: Early-mid Miocene erosion rates measured in pre-Dead Sea rift Hazeva River using cosmogenic 21Ne in fluvial chert pebbles

Ben-Israel et al. present 10 new in situ-produced 21Ne concentrations from quartz and chert material in the pre-Dead Sea rift Hazewa River located in southern Israel. Where possible in situ-produced 10Be and 26Al concentrations are provided for the same sample material. The data from sand, pebbles, and nodules is used to determine Early-mid Miocene erosion rates.

General comments:

The manuscript is generally well written and reads well. However, there are several

weak points in the manuscript which need to be clarified and improved:

1. The interpretations of the data are relatively strong given the amount of available data. As the manuscript stands right now, it is not clear to me if the given interpretations of the results would stand if more data is available. For instance, the nuclide concentrations of the two chert nodules do not agree with continuous erosion of a landscape. In order to investigate the problem more samples should be analyzed. However, knowing that this is easy to say and that cosmogenic nuclide analysis are expensive, a request for more data is not at the right place. Instead a request to tone down the interpretation is made.

2. The method section needs to be set-up in a logical and rigorous way (see detailed comments below). The different methods applied need to be described in more detail. The calculations performed and parameters used explained. In general, the order of the presented information could be rearranged to make understanding easier. Concise wording and details in tables and figures are needed (see comments below).

3. The interpretation of the data needs to be more rigorous. For instance, the nuclide concentrations could be normalized to the same altitude. This would make a comparison more meaningful. A chert nodule at 1000 m above sea level subjected to slow erosion rates is expected to have high nuclide concentrations. Concentrations and erosion rates could also be investigated with "banana plots" for the three different nuclides (e.g., Ivy-Ochs and Kober, 2008). Such plots would help to visualize the data presented in Table 1. In addition, the use of erosion rate in combination with the integration time could be helpful for the reader.

Unfortunately, I am not an expert in the measurements of cosmogenic 21Ne. Therefore, the quality and reliability of the presented measurements are not assessed in this review. This is hopefully addressed by another reviewer. For instance, it is not clear to me what happens to 21Ne during sample preparation when leached at 150°C (see line 118-9)? This question comes up as just some lines above (86-9) the closure temperature for 21Ne in quartz is mentioned to be 90 - 100°C. As diffusion is a slow process, the used sample preparation may be valid. Clarification is needed.

Detailed comments:

Abstract

L11-14 The abstract jumps to much into details which are not relevant for the abstract. It would make more sense to give the reader a reason why this study was done and what the goal is.

L15-18 What do you mean by "modern erosion rates"? Do these rates not integrate over hundred thousand of years? Please clarify.

L16-17 Would it make more sense to first report nuclide concentrations and then move to erosion rates?

L20-21 Is bedrock erosion equal to modern erosion?

L21-23 Long sentence not easy to understand. What does the even mean? Are the currently eroding chert nodules not used for the bedrock erosin? Needs clarification.

L24-25 From what material are the "rates calculated today"? And what does it mean "rates calculated today"? As mentioned in the general comments, it might be better to compare (normalized) concentrations rather than erosion rates.

1 Introduction

L37-40 This sentence is not easy to understand for a reader not familiar to the technique of in situ-produced cosmogenic nuclides. Can this sentence be extended? What is the limitation of the method due to the half-lives?

L44 "...parts of the..."

L46-48 The introduction comes to a quick end. An outline of the study set-up and a hypothesis to be tested would be helpful for the reader. What kind of samples do you

analyze with what method? What are the questions to be answered with this study?

2 Geological Background

L49 This chapter would gain a lot if called "Study Area" and start with a geographic description of today and what deposits are present. Then move over to the geologic background. Changing this order would request to change Fig. 2 to Fig. 1 and vice versa.

3 Methodology and Analytical Procedures

L76-89 This first section could be labeled 3.1 Cosmogenic method. It could contain the existing information, but also explain the method of catchment-wide erosion rates. In addition, it could contain further explanations of possible problems faced. Not all readers know the caveats of the cosmogenic nuclide methods (e.g., radioactive versus stable nuclides). Would it make sense to discuss here the possible influence of the transport distance and time on radioactive and stable cosmogenic nuclides?

L90 Now start here: 3.2 Sampling Strategy?

L134 How is the chemistry blank correction performed? What are the values used for erosion rate calculations?

4 Results

L143-5 This sentence needs clarification. What is exactly 21Neex?

L160-1 What is about differences in 21Neex in chert and quartz samples? Could this have an influence on the 21Necos concentration?

L178-9 Difficult sentence to understand. Please clarify.

5 Discussion

L193 What are the parameters used for these calculations?

L210 The use of "Modern" is misleading.

L211 On what are the calculations of the erosion rates based? Sea level-high latitude production rate, production rate scaling, etc.? This information should be given in the methods.

L215 Are these reported values correct? Please cross-check.

Figures

Fig. S5 - S11: Please label x- and y axes.

Tables

Table 1: Please cross-check units.

---

## Referee Comment (RC3) · Marissa Tremblay (Referee) · 17 Nov 2019

In this discussion paper, Ben-Israel et al. present new neon isotope measurements from quartz sand and chert pebbles deposited by the Hazeva River, which drained the Arabian Peninsula during Miocene times, as well as from modern eroding outcrops where the chert pebbles were likely sourced. The authors compare apparent erosion rates calculated from cosmogenic 21Ne concentrations at both sites, and conclude that the erosion rates recorded by the Miocene fluvial deposits are higher than those in the modern. They attribute higher Miocene erosion rates to higher uplift rates and a wetter climate of the Arabian Peninsula at that time.

Major comments:

In general, I think the approach taken in this paper to quantify paleo-erosion rates is exciting. However, I am concerned that the uncertainties in the paleo-erosion rates the authors calculate are underestimated, and that therefore the conclusions about higher erosion rates during the Miocene are overstated. Specifically, the authors assume that the elevation during Miocene production of cosmogenic 21Ne was 1000 km. It is unclear to me if this is the assumed elevation of the source of the chert pebbles, and that the authors then assume that the majority of cosmogenic 21Ne production occurs prior to sediment transport? Or is 1000 km accounting for sediment transport and meant to be representative of some catchment-integrated value between where the pebbles were sourced and deposited? Furthermore, it appears that the authors do not give this paleo-elevation any uncertainty in their calculation of exposure times or minimum erosion rates. I suspect that if the authors incorporate a reasonable elevation uncertainty of something like ïĆś 500 m, that their paleo-erosion rates will overlap entirely with their modern erosion rates. Because the choice of a paleo-elevation has such a large effect on the calculated paleo-erosion rates, there needs to be (1) a more detailed explanation of what the paleo-elevation the authors use represents, and (2) an uncertainty associated with this paleo-elevation incorporated into the calculated paleo-erosion rates.

I am also wondering if the authors need to be worried about neon diffusion with their analysis chert pebbles. They cite a neon closure temperature for quartz reported by Shuster and Farley (2005), but this was calculated for a 500 micron quartz grain. What is controlling the diffusion lengthscale in these chert pebbles, and what is the typical grain size in the chert pebbles analyzed? If you look at Figure 4 from Shuster and Farley (2005), for a 200 micron-diameter quartz grain (e.g., log radius of -1), you would expect significant diffusive loss of neon on 100 ka timescales and at temperatures of 60-70 °C. These seem like high temperatures, but in the Arabian Peninsula air temperatures regularly exceed 40 °C in the summer months and rock temperatures can

certainly be in the range of 60-70 °C (e.g., McFadden et al., 2005). Additionally, the fact that the authors observe lower degassing temperatures in the laboratory for the chert samples than they do for the quartz samples suggest that the chert has a lower thermal sensitivity. Altogether, this makes me think that diffusion might be contributing to the observation that the chert pebbles have lower cosmogenic 21Ne concentrations than the quartz sands. Given this, I think a discussion of the potential role of neon diffusion needs to be added to the text.

Minor comments:

Figure 1 should have a box indicating the location of figure 2A.

Figure 2 should also include field photos of the modern in situ sampling sites of chert nodules.

Lines 204-206: This sentence is awkwardly worded, and I'm not sure I fully understand I follow the logic.

Lines 255-268: The calculated paleo-erosion rates overlap with the upper end of the modern erosion rates, even without my concerns about the paleo-elevation uncertainty being addressed. This sentence (and similar statements elsewhere) overstates the significance of the authors findings. I think it would be more appropriate to say that the calculated paleo-erosion rates allow for the possibility of higher erosion rates in the Miocene.

Lines 281-283: Here and elsewhere, do you mean rock uplift or surface uplift?

Supplement: There needs to be some text explaining what is provided in each of the Excel spreadsheet tabs as well as a caption provide for each of the supplemental figures. It's not obvious to me why you need all of the different neon three isotope plots and why they are in the order that they are presented in. This could be cleaned up by having one three isotope plot for the Miocene samples and one for the modern samples, and using different symbol shapes to represent the different temperature steps.

[Figure]

Sincerely,

Marissa Tremblay Purdue University November 17, 2019

New references cited: McFadden, L.D., Eppes, M.C., Gillespie, A.R., and Hallet, B., 2005, Physical weathering in arid landscapes due to diurnal variation in the direction of solar heating: Geological Society of America Bulletin, v. 117, p. 161, doi:10.1130/B25508.1.
* * *

---

## Referee Comment (RC4) · Marissa Tremblay (Referee) · 21 Nov 2019

This is just a minor clarification of my original review, in light of the response written by Ben-Israel et al. I appreciate that the Miocene samples were sufficiently buried to not worry about thermal loss during burial or during more recent exposure. My concern about neon diffusion was in regard to diffusion that might have occurred during Miocene erosion and sediment transport, prior to burial. The timescales of Miocene exposure to cosmic rays the authors calculate are on the order of 10^5 years, and these are the timescales over which there could be nontrivial diffusive loss of neon at 60-70 C, depending on grain size.

[Figure]

I appreciate that this could be a tricky issue to tackle, because you want to avoid making a circular argument (i.e., the climate during the Miocene was wetter/had less extreme temperatures, so we don't have to worry about diffusion, and not worrying about diffusion gives us a certain range of erosion rates, which indicates that the climate was different). However, I think an attempt to address the potential for neon diffusion during Miocene in the text is worthwhile.

Sincerely,

Marissa Tremblay Purdue University November 21, 2019

---

## Author Comment (AC1) · 21 Nov 2019

We thank Dr. Schildgen for the insightful and thought-provoking comments. We will incorporate many of the suggested changes and modify the manuscript to address the concerns raised. Mainly, the concerns raised are regarding the calculation and interpretation of the reported erosion rates.

The first comment is regarding the paleo-elevation of the Arabian Peninsula during the early-mid Miocene and the consequential production rate of cosmogenic nuclides during the deposition of the Hazeva formation. While we somewhat address this issue in the manuscript, we agree that the references cited do not provide clear evidence to con-

strain the timing of uplifting of the western half of the Arabian Peninsula and can only determine that it was uplifted to its current elevation over the last 30 Myr (see Feinstein et al., 2013; Willson et al., 2014). We will now include an additional calculation based on a moderate 0.5% gradient westward from a base level at the Mediterranean margin and over a ~200 km distance to the location of the currently exposed source rock. This simple calculation supports an elevation of ~1 km above sea level. Additionally, based on a comment made by the third reviewer (Marissa Tremblay), we now consider a range of possible elevations between 500-1000 m and the ensuing production and erosion rates. We will alter the discussion, conclusions, and abstract accordingly in the revised manuscript.

The second concern is about the concluded differences between the Miocene and modern erosion rates. While it is true that both rates are relatively slow, it is important to remember that we are not comparing apples to apples – the modern rates are calculated for slowly eroding exposed surfaces, all that these samples have undergone is exposure and weathering from the source rock. In contrast, the Miocene samples were eroded, transported in the fluvial system, and deposited all the while exposed/partially exposed to cosmic rays. Therefore, we consider the calculated paleo-erosion rates to be a minimum calculation. Nevertheless, while we consider the difference between modern and paleo-erosion rates to be meaningful, we agree that the Miocene erosion rates are not comparable to rates measured in rapidly eroding locations, and we will address this in the revised discussion and conclusions.

The last comment made regarding erosion rates has to do with the small number of samples collected from currently eroding surfaces, i.e., in the Central Jordanian Plateau. Unfortunately, this area is very difficult to access currently, and therefore, we can only use the few samples collected there previously. However, we will now include in the manuscript data from several previous studies of modern erosion rates from locations across the Negev Desert, demonstrating the reported erosion rates are representative for this region (e.g., Boroda et al., 2014; Fruchter et al., 2011; Matmon

et al., 2009; Matmon and Zilberman, 2016; Matmon et al., 2016). In addition to these concerns, two suggestions for the improvement of the manuscript were made, one to do with emphasizing the difference between the quartz sand and chert pebbles when it comes to inherited cosmogenic 21Ne in sediments. Thank you for this insight, we will now further discuss the different types of detrital material and how to interpret the measured cosmogenic 21Ne in them for calculating rates of surface processes. Finally, there are several line-specific comments, edits, and questions. We will address each one in the revised manuscript.

——————————— Boroda, R., Matmon, A., Amit, R., Haviv, I., Arnold, M., Aumaître, G., Bourlès, D.L., Keddadouche, K., Eyal, Y., and Enzel, Y., 2014, Evolution and degradation of flat-top mesas in the hyper-arid Negev, Israel revealed from 10 Be cosmogenic nuclides: Earth Surface Processes and Landforms, v. 1621, p. 1611-1621, doi: 10.1002/esp.3551. Feinstein, S., Eyal, M., Kohn, B.P., Steckler, M.S., Ibrahim, K.M., Moh'd, B.K., and Tian, Y., 2013, Uplift and denudation history of the eastern Dead Sea rift flank, SW Jordan: Evidence from apatite fission track thermochronometry: Tectonics, v. 32, p. 1513–1528. Fruchter, N., Matmon, A., Avni, Y., and Fink, D., 2011, Revealing sediment sources, mixing, and transport during erosional crater evolution in the hyperarid Negev Desert, Israel: Geomorphology, v. 134, p. 363–377, doi: 10.1016/J.GEOMORPH.2011.07.011. Matmon, A., Simhai, O., Amit, R., Haviv, I., Porat, N., McDonald, E., Benedetti, L., and Finkel, R., 2009, Desert pavement-coated surfaces in extreme deserts present the longest-lived landforms on Earth: Geological Society of America Bulletin, v. 121, p. 688–697, doi: 10.1130/B26422.1. Matmon, A., and Zilberman, E., 2016, Landscape Evolution Along the Dead Sea Fault and Its Margins, in Enzel, Y. and Bar-Yosef, O. eds., Quaternary of the Levant, Cambridge University Press, p. 771. Matmon, A., Elfassi, S., Hidy, A., Geller, Y., and Porat, N., 2016, Controls on aggradation and incision in the NE Negev, Israel, since the middle Pleistocene: Geomorphology, v. 261, p. 132–146, doi: 10.1016/j.geomorph.2016.02.020. Morag, N., Haviv, I., Eyal, M., Kohn, B.P., and Feinstein, S., 2019, Early flank uplift along the Suez Rift: Implications for the role of mantle

plumes and the onset of the Dead Sea Transform: Earth and Planetary Science Letters, v. 516, p. 56–65, doi: 10.1016/j.epsl.2019.03.002. Wilson, J.W.P., Roberts, G.G., Hoggard, M.J., and White, N.J., 2014, Cenozoic epeirogeny of the Arabian Peninsula from drainage modeling: Geochemistry, Geophysics, Geosystems, v. 15, p. 3723–3761, doi: 10.1002/2014GC005283.
* * *
**ESurfD**
* * *

---

## Author Comment (AC2) · 21 Nov 2019

We thank reviewer #2 for her/his time and for the constructive and helpful comments. We will incorporate many of the suggested changes and modify the manuscript to address the concerns raised. Generally, there were three major comments made, some of which were similarly made by the first reviewer (Taylor Schildgen) and we include our responses to these concerns below.

The first comment is regarding the small number of modern samples analyzed. As the first reviewer made a similar point, I include here the answer given there. Unfortunately, the source area for chert nodules is very difficult to currently access and therefore

[Figure]

we can only use the few samples collected there previously. However, we will now include in the manuscript data from several previous studies of modern erosion rates from locations across the Negev Desert, demonstrating the reported erosion rates are representative of this region (e.g., Boroda et al., 2014; Fruchter et al., 2011; Matmon et al., 2009; Matmon and Zilberman, 2016; Matmon et al., 2016).

The second main comment is concerning the interpretation of the measured data and the uncertainty regarding the elevation of the samples during the Miocene, as well as a suggestion to use 'banana plots' to investigate the slowly eroding samples. The first reviewer also brought up the concerns about the elevation during the Miocene and I include here the answer given. While we somewhat address this issue in the manuscript, the references cited do not provide clear evidence to constrain the timing of uplifting of the western half of the Arabian Peninsula and can only determine that it was uplifted to its current elevation over the last 30 Myr (see Feinstein et al., 2013; Willson et al., 2014). We now include an additional calculation based on a 0.5% moderate gradient westward from a base level at the Mediterranean margin and over a ∼200 km distance to the location of the currently exposed source rock. This simple calculation supports an elevation of ∼1 km above sea level. Additionally, based on a comment made by the third reviewer (Marissa Tremblay), we now consider a range of possible elevations between 500-1000 m and the ensuing production and erosion rates. We will alter the discussion, conclusions, and abstract accordingly in the revised manuscript.

Regarding the suggestion made to include a more familiar tool for presenting burial time at different erosion rates, unfortunately applying this type of diagram will not help to better explain the reported data. When examining samples that have been buried for extensive periods, the measured concentrations of cosmogenic 26Al and 10Be do not represent burial time, as the samples have reached a steady-state determined by post-burial (muonic) production. We did include a figure that presents changes in steady-state concentration with time (see fig. 4). Furthermore, when comparing the cosmogenic 21Ne concentration between the two samples, this is also not a straight-

forward comparison due to the different sedimentary histories of the different types of samples. While the currently eroding chert nodules have only undergone exposure and weathering from the source rock. In contrast, the Miocene samples were eroded, transported in the fluvial system, and deposited all the while exposed/partially exposed to cosmic rays. Therefore, a 'simple comparison' cannot be made. Nevertheless, we will do our best to explain this point more clearly in the revised manuscript.

Regarding the final major comment, regarding possible Ne diffusion during sample preparation, as the manuscript suggests, 80-90°C is the closure temperature of Ne in quartz over geological timescales. The fractional loss of Ne due to diffusion over a 1.5-hour timeframe is insignificant (<0.1%). We realize that cosmogenic Ne experts are few and far between and we will rephrase this part in the manuscript to clear up any confusion.

Lastly, regarding the detailed minor comments, edits, and questions. We will address each one in the revised manuscript.

———————————— Boroda, R., Matmon, A., Amit, R., Haviv, I., Arnold, M., Aumaître, G., Bourlès, D.L., Keddadouche, K., Eyal, Y., and Enzel, Y., 2014, Evolution and degradation of flat-top mesas in the hyper-arid Negev, Israel revealed from 10 Be cosmogenic nuclides: Earth Surface Processes and Landforms, v. 1621, p. 1611-1621, doi: 10.1002/esp.3551. Feinstein, S., Eyal, M., Kohn, B.P., Steckler, M.S., Ibrahim, K.M., Moh'd, B.K., and Tian, Y., 2013, Uplift and denudation history of the eastern Dead Sea rift flank, SW Jordan: Evidence from apatite fission track thermochronometry: Tectonics, v. 32, p. 1513–1528. Fruchter, N., Matmon, A., Avni, Y., and Fink, D., 2011, Revealing sediment sources, mixing, and transport during erosional crater evolution in the hyperarid Negev Desert, Israel: Geomorphology, v. 134, p. 363–377, doi: 10.1016/J.GEOMORPH.2011.07.011. Matmon, A., Simhai, O., Amit, R., Haviv, I., Porat, N., McDonald, E., Benedetti, L., and Finkel, R., 2009, Desert pavement-coated surfaces in extreme deserts present the longest-lived landforms on Earth: Geological Society of America Bulletin, v. 121, p. 688–697, doi:

10.1130/B26422.1. Matmon, A., and Zilberman, E., 2016, Landscape Evolution Along the Dead Sea Fault and Its Margins, in Enzel, Y. and Bar-Yosef, O. eds., Quaternary of the Levant, Cambridge University Press, p. 771. Matmon, A., Elfassi, S., Hidy, A., Geller, Y., and Porat, N., 2016, Controls on aggradation and incision in the NE Negev, Israel, since the middle Pleistocene: Geomorphology, v. 261, p. 132–146, doi: 10.1016/j.geomorph.2016.02.020. Morag, N., Haviv, I., Eyal, M., Kohn, B.P., and Feinstein, S., 2019, Early flank uplift along the Suez Rift: Implications for the role of mantle plumes and the onset of the Dead Sea Transform: Earth and Planetary Science Letters, v. 516, p. 56–65, doi: 10.1016/j.epsl.2019.03.002. Wilson, J.W.P., Roberts, G.G., Hoggard, M.J., and White, N.J., 2014, Cenozoic epeirogeny of the Arabian Peninsula from drainage modeling: Geochemistry, Geophysics, Geosystems, v. 15, p. 3723–3761, doi: 10.1002/2014GC005283.

---

## Author Comment (AC3) · 21 Nov 2019

We thank Dr. Tremblay for her time and for the constructive comments and her insightful review. We have incorporated many of the suggested changes and modified the manuscript to address the concerns raised. Of the major comments made, some have been addressed on our response to the first reviewer (Taylor Schildgen), and we include our responses to these concerns below.

The first major concern is regarding the calculation and interpretation of the reported erosion rates and more specifically, suggesting that the elevation used ( $\sim$ 1 km) is not representative and a catchment-integrated value should be used for calculations of

erosion rates. This is a very good point and incorporating this with the comment made by the first reviewer, we now consider a range of possible elevations between 500-1000 m and the ensuing production and erosion rates. We will alter the discussion, conclusions, and abstract accordingly in the revised manuscript.

The other major comment is regarding Ne diffusion from the Miocene samples during their burial in the Negev Desert over the past ~18 Myr. An interesting point is made here both regarding the 'typical grain size' in the chert pebbles and the temperatures reached during their burial in the Negev Desert. While it is very true that air temperatures get very high in the Negev and the temperatures of exposed dark rock (such as cherts) can get up to 60-70°C and very possibly even higher. However, surface heating only effects the few top cm's and the Miocene samples presented in the manuscript were not collected from the surface. Miocene samples are buried continuously bellow tens of meters of sediment so they are not exposed to direct sunlight meaning temperatures are much cooler. In addition, these samples were collected either from an active quarry or from a shielded (and well-shaded) outcrop. We, therefore, have no reason to suspect that any significant diffusion occurred\*. We will do our best to explain this point clearly in the revised manuscript.

As for the question raised regarding the chert mineral size diameter, we do not know with certainty whether or by how much the kinetic parameters of chert differ from that the ones measured for quartz. We agree that this point and the possible implications should be further discussed and we intend to do that in the revised manuscript.

Finally, there are several minor comments, edits, and questions. We will address each one in the revised manuscript.

**ESurfD**
weathering (see darker fragment in the center of Fig. 1). This chert has nothing to do with the samples presented in the submitted manuscript (!) - it's just a nice example of surface weathering of chert. For comparison, we also attach a photo of Miocene pebbles from Arad Quarry. Notice its pristine condition, this pebble did not experience direct exposure to radiation at the surface (Fig. 2).

**ESurfD**

---

## Author Comment (AC4) · 21 Nov 2019

Thanks for the clarification. This is indeed an interesting point and one that we will discuss with regard to all samples in the revised manuscript.

As a general note, I think this type of problem should mostly be considered in cases where desert surfaces are dated for extended exposure. In the case of fluvial sediment, as temperatures drop dramatically even a few centimeters below the surface (unlike production rate that only decreases a little), it seems impossible to make an estimation for diffusion. For fluvial samples, we have no way of knowing whether they were actually at the surface or at 5 to tens cm depth, which is most likely the case

during most of the fluvial transport.

I agree that this is a worthwhile discussion and one that should take place in this manuscript.
* * *

---

## Author Response (AR1)

**Response to referees – ESurf-2019-54**

Dear Dr. Masteller,

Thank you and the handling editor for the editing advice and to the three reviewers for their time and for the constructive and insightful reviews. We have incorporated many of the suggested changes and modified the manuscript to address the concerns raised, and we are pleased to include the revised manuscript.

Please find below are our replies to different comments by each referee in blue. To summarize, we have made significant changes to the discussion section so it now address the issues raised regarding the calculations of paleo-production rates and the derived erosion rates. We now focus on exposure times comparing between the modern and Miocene cherts, and consider these differences as they pertain to erosion rates. We have also added an additional table (Table 2), which includes exposure times and erosion rates calculated using paleo-elevations that range 500-1000 m.a.s.l in order to make comparison easier for the reader. We hope that the revisions made to the discussion as well as other sections will now make the manuscript clearer and easier to follow.

Please find attached below the revised manuscript with our additions and replacements displayed in tracked changes mode. Page and line numbers refer to the revised version of the manuscript in 'Simple Markup' setting.

We believe that our manuscript is much improved and we hope it is now ready for publication in ESurf. Please contact me for any further questions or information.

Sincerely,

Michal Ben-Israel

Michal Ben-Israel

(on behalf of all the authors)

Response to Reviewer no. 1: Taylor Schildgen

**General comments:**

Ben-Israel and co-authors analyze stable (21Ne) and unstable (10Be, 26Al) cosmogenic nuclides in detrital Miocene sediments from the NW Arabian plateau to calculate paleo-erosion rates and compare these to "modern" rates obtained from bedrock outcrops. They interpret an approximate 2-fold difference in concentrations to result from erosion rates that were 2x faster during the early-mid Miocene, which could be reasonable considering evidence for a wetter climate in the region at that time. On the positive side, this work illustrates the unique ability of the stable cosmogenic nuclide 21Ne to record erosion rates averaged over relatively short time intervals (if we can consider 100s of kyr short) in ancient, well-shielded sedimentary deposits; such information cannot be obtained with 10Be or 26Al. The authors are careful in their consideration of various potential complications of their data – post-burial deposition, changes in elevation through time, and how different types of detrital material (quartz sand v. chert pebbles) could have experienced very different pathways to the final deposition site. I also really like the use of different types of detrital material to assess the possibility of 21Ne inheritance.

We thank prof. Schieldgen for these comments.

But I see several areas that require improvement. Most concerning for me is the discussion of the uplift-history constraints, which I don't find very convincing, but are critical for the final interpretations, as the difference in measured 21Ne concentrations between modern and mid-Miocene samples can be explained either by a change in elevation through time or a change in erosion rates.

This is a very good point and one that has been similarly made by the other reviewers. While we consider this uncertainty in the original manuscript, we agree that the discussion into the interpreted paleo-erosion rates should have been done more circumspectly. We now include additional discussion into the Miocene paleo-elevation and the calculated production rates. In addition to the stratigraphic and cooling ages evidence we now also calculate the possible elevation using a moderate continental slope. Furthermore, to account for the uncertainty in paleo-elevation as well as basin scaled paleo-production rates, we now consider an elevation uncertainty range of 500-1000 m.a.s.l (see lines 250-289, and table 2).

But even if there has not been a change in elevation, the difference in erosion rates reported (2-4 mm/kyr v. 4-12 mm/kyr) is not huge, and those mid-Miocene rates are still pretty darn slow. Is this really a story about how climate affects erosion rates, or could the conclusion be that it doesn't affect them all that much?

This is a valid point, and while it is true that the differences between modern and paleo-erosion rates calculated are relatively slow, it is important to note that the comparison made is not straighforward. The modern rates represent only erosion from bedrock, while the Miocene erosion rates represent both erosion from bedrock and transport in the Miocene river. There is no way of evaluating what were the actual bedrock erosion rates during the Miocene, only that they must have been faster. This point was is now made more clearly with the comparison between exposure times throughout most of the disscusion

with erosion rates mentioned only in the final part of the manuscript. As mentioned, in these (currently) hyper-arid environment, even small changes to erosion rates are significant. See the revised disscusion (lines 333-343) and conclusions (lines 377-381).

I'm also concerned by the small number of samples obtained from the outcrops (only two), and the possibility that the rates reported are not representative of modern rates (often spot samples from outcrops lead to a wide range of erosion-rate estimates). Are there any other modern erosion rates that have been reported that can be used to corroborate the results presented here? Is there a reason a modern erosion-rate estimate wasn't made from modern detrital sands in the region, even if that rate would not be from exactly the same drainage area as the early-mid Miocene samples?

Unfortunately, the area where A. Matmon and Y. Avni collected the modern chert nodules is very difficult to access these days, and it is not possible to collect any more samples. However, as we point to in the manuscript, there has is an extensive body of work looking into rates of erosion of chert and quartzite surfaces in the hyper-arid Negev desert (e.g., Boroda et al., 2014; Fruchter et al., 2011; Matmon et al., 2009; Matmon et al., 2016; Matmon and Zilberman, 2017). We now references rates from these studies (see lines 307-309).

I think the difference between the detrital quartz sand and detrital chert pebbles can be better emphasized in the final interpretations/conclusions of the paper. It seems that the quartz-sand results are not considered in the final interpretations due to the possibility that the quartz experienced multiple periods of deposition and exposure prior to the last deposition, hence it contains inherited 21Ne (if that's the reason that the erosion rates for the quartz samples are not reported, the authors should state that explicitly rather than leaving it for the readers to infer; still I think the erosion rates should be reported). But rather than making it seem as if those samples were just a waste of time, it could be helpful to emphasize how in recycled sediments, inherited 21Ne can be a real problem. I like the approach here of measuring different types of detrital material to assess this possibility! That could be highlighted, rather than hidden.

That was by no means our intention. Inherited cosmogenic 21Ne is one of the disadvantages of using cosmogenic 21Ne and we include this limitation throughout the manuscript and in the conclusion section (see lines 373-376).

Finally, I think the introductory paragraph can be improved; several of my line-specific comments refer to my confusion about where the paper is going just in the first several sentences.

We have now revised the introduction paragraph (see lines 28-54).

**Line-specific comments:**

l. 36-37: Older landscapes are transient? Odd wording. Also, this sentence doesn't really follow from the previous ones. You've discussed river systems and sediment archives, now we're on to preservation of landscapes themselves? Be more precise and focused.

We have changed the phrasing in this section. See lines 35-46.

l. 38: Okay, so the focus is on quantifying erosion rates from surfaces? This is not easy to follow.

See previous comment.

l. 41: If the focus is on erosion, don't change the terminology here to "surface processes", as that encompasses much more than just erosion.

See previous comment.

l. 43: Now you've explained that the focus is on sedimentary deposits, not slowly eroding surfaces. I suggest rewriting this whole paragraph with a clearer focus on what information you want to give to the reader. What is the main problem, why is it difficult to address, how are you going to do it?

See previous comment.

l. 51: Wouldn't it be the other way around, i.e., the Afar plume leads to magmatic events, and maybe even influenced tectonics?

We have changed the phrasing in this section. See lines 56-65.

l. 67-69: This means that the deposition associated with the river started prior to ca. 20 Ma? Or do you interpret only the upper part of the Hazeva Formation to be associated with the river, meaning that fluvial deposition started after 20 Ma? Please clarify.

The sediments in lower part of the Hazeva formation are local and were not deposited by the Hazeva River (unlike the upper part of the section). We have now clarified this in the text (see lines 79-92).

l. 99-101: How is this history of the quartz sand known? If this history is going to be important for explaining differences between 21Ne measured in quartz vs. chert, then a fuller explanation is needed. One potential worry with quartz sand is that it could be aeolian in origin; can this be ruled out?

The history, provenance and stratigraphy of the Hazeva formation was previously researched and is reported by Calvo and Bartov (2001) and Zilberman and Calvo (2013) and is beyond the scope of this manuscript. While an eolian history cannot be ruled out, it is not likely. There is no reason to assume that sediment eroded from bedrock and deposited in a fluvial environment is of eolian source. We now try to make this clearer in the text (see lines 76-81).

What size fraction of sand was processed?

The grain size fraction analyzed is now reported in lines 134-135.

l. 90-105: I suggest moving this paragraph to the geological setting, as it provides the geological context for the samples collected.

This paragraph has been moved and modified. See lines 75-95.

l. 109: How deeply shielded were the collected samples? Deeply enough to rule out post-depositional 21Ne production?

This question is thoroughly discussed in the discussion section. We now refer to this in the text (see line 116).

l. 113: I suggest "accumulated cosmogenic nuclides only during exhumation", as the samples that experienced the full sedimentary cycle also accumulated nuclides during exhumation.

We accept this correction. See line 127.

l. 156-157: Is this because you assume the U and Th are equally distributed throughout the rock? Is that a reasonable assumption?

U and Th are most likely found in inclusions within the crystal lattice. The U/Th concentration in the cherts is determined by the age of the rock and the environmental conditions during its formation (or later metamorphosis). It is reasonable to assume that is would the same or similar for all the presented cherts as they all share the same lithology.

l. 212-217: Don't assume that your readers remember that EJC5 and EJC3 were the surface samples of from the "in situ" outcrops, remind us.

Maybe this comment can be clarified as the first part of the paragraph describes exactly that (see lines 246-259).

l. 214-215: This detail concerning the scaling of production rates belongs in the methods, not the discussion.

We now include these details in the methods section (see section 3.3, lines 152-156).

l. 223-224: I don't see the added value of reporting equivalent exposure times (if that is what is meant by "simple exposure time"), given that you are mainly interpreting the measured concentrations in terms of erosion rates. Or is the goal to give readers a sense of the averaging timescale of these erosion rates? If the latter, I suggest rewording to make this clear.

The revised manuscript now focuses on exposure times (only later do we discuss the inferred erosion rates). The calculated exposure times are explained in lines 246-249.

l. 224-226: It is important, but why? I can make a guess, but it would better if you explain.

We now explain this, see lines 310-312.

l. 230-231: Could you please briefly remind me what those differences in concentrations are?

This section has now been revised and we no longer discuss 21Ne concentrations but exposure ages, which are reported in lines 291-293 and in Table 2.

l. 237-239: A bigger overview map that includes the Suez rift in addition to the all the other relevant sites mentioned here would be very helpful.

See our revisions to figure 2, and our comments to this there.

l. 237-242: These uplift constraints are crucial for your interpretation of whether or not there the Miocene samples show a faster erosion rate compared to today or reflect a similar erosion rate with a lower nuclide paleo-production rate. Given their importance, some more details on these uplift constraints would be very helpful. Although I have not checked each of the references in detail, I disagree with how you have referenced the Wilson et al. (2014) interpretations. Despite many reasons why these interpretations of uplift histories from river profiles should be considered suspect, their interpretation for your field area is that most of the modern elevation gain occurred since 20 Ma, and it looks like more than half of that is since 10 Ma (see their Fig. 17). For that reason, I don't agree at all with your statement that it is reasonable to presume that the western flank of the Arabian Peninsula (or the NW edge, corresponding to your field area) reached its current elevation prior to the initiation of the Hazeva fluvial system at ca. 18 Ma.

As is mentioned in our answers to the general comments, we agree with this comment. Our discussion regarding the elevation of the Arabian Plateau during the Miocene should have been more thorough. We now provide additional evidence to support our assumption of the paleo-elevation as well as consider a significant uncertainty for this (see lines 250-289). Regarding Wilson et al. (2014), both in figure 17 and in figure 21, the central part of the Arabian plate appears to be stable ~20 Myr ago (unlike the southwestern tip).

l. 255-258: I can guess why you do not mention erosion rates from the quartz sand – because you suspect it has inherited 21Ne – but it seems like an oversight. I suggest to not "hide" those results, but rather emphasize how recycling of quartz sand can lead to incorrect results.

As is mentioned in our answers to the general comments, hiding this aspect was not at all our intention. We refer to the possible inheritance in quartz throughout the manuscript (including the conclusions section). The discussion about the possible effects of it is slightly beyond the scope of this manuscript but is referenced to (see Ben-Israel et al., 2018 for further reading).

l. 281-287: Mostly I've been able to work out myself whether you are referring to surface uplift or rock uplift up until now, but in this section in particular I cannot follow your meaning. In this paragraph, and ideally throughout the manuscript, specify which one you are referring to.

As is mentioned in our answers to the general comments, hiding this aspect was not at all our intention. We refer to the possible inheritance in quartz throughout the manuscript (including the conclusions section). The discussion about the possible effects of it is slightly beyond the scope of this manuscript but is referenced to (see Ben-Israel et al., 2018 for further reading).

l. 285: But this is not an accurate representation of the uplift history for your field area as interpreted by Wilson et al. (2014).

This section has now been revised – see out previous comments and lines 344-351.

l. 288-300: This evidence for a wetter early to mid Miocene climate seems reasonable, and I agree that such a climate would likely erode faster than low-relief, hyperarid landscapes. But the mid-Miocene erosion rates reported here, which might be considered maximum rates given the uncertainty in the

paleo-elevation, still seem very slow. How do rates of 4 to 12 mm/kyr compare with erosion rates measured from similar environments today? (Incidentally, I realize I'm assuming that the landscape relief is relatively low, but it would be helpful to actually show a slope/relief map to see whether or not

that's the case).

This section has now been revised – see out previous comments and lines 332-343.

Figure 2: As mentioned above, a broader overview map would be very helpful. In 2B, is that a person near the bottom? Highlighting or circling him/her in some way would make it easier to understand the scale of this photo. Likewise, in 2C, is that a dog?

Figure 2 has been revised and now includes a more zoomed overview map with Red Sea marked on it. It additionally now includes clear marking of a human (Dr. Avni) in 2B and dog (Kara) in 2C.

Figure 3: Given the overall focus on erosion rates, I find it odd that the calculated erosion rates are not shown in this figure. Why not use those instead of the effective exposure ages?

With the addition of Table 2, figure 3 no longer includes exposure ages (or erosion rates).

**Editorial comments:**

l. 35: always specify what you mean after "this", e.g., this lack of information?

Corrected. See line 38.

l. 64: "comprise" rather than "compose"

Corrected. See line 78.

l. 71: I'd suggest "disruption" rather than "dismantlement"

Corrected. See line 71.

l. 81: What "cycles" are you referring to here? Okay, you clarify it in the next sentence, but please instead clarify at your first instance of using this term.

Corrected. See lines 99-102.

l. 86: This what should hold true?

This section has been revised. See lines 192-202.

l. 96: comprising, not composing.

Corrected. See line 88.

l. 210: Please refer to "denudation" or "erosion" rates throughout, not "rates of surface processes", which is unnecessarily vague.

We have corrected this in the referenced line (245) and throughout the manuscript.

l. 221: lots of needless words here, please shorten to "erosion rates between 1 and 5 mm/kyr"

This section has been revised. See lines 307-309.

**Response to Reviewer no. 2: Anonymous Referee**

Ben-Israel et al. present 10 new in situ-produced 21Ne concentrations from quartz and chert material in the pre-Dead Sea rift Hazewa River located in southern Israel. Where possible in situ-produced 10Be and 26Al concentrations are provided for the same sample material. The data from sand, pebbles, and nodules is used to determine Early-mid Miocene erosion rates.

**General comments:**

The manuscript is generally well written and reads well. However, there are several weak points in the manuscript which need to be clarified and improved:

We thank the reviwer for these comments.

1. The interpretations of the data are relatively strong given the amount of available data. As the manuscript stands right now, it is not clear to me if the given interpretations of the results would stand if more data is available. For instance, the nuclide concentrations of the two chert nodules do not agree with continuous erosion of a landscape. In order to investigate the problem more samples should be analyzed. However, knowing that this is easy to say and that cosmogenic nuclide analysis are expensive, a request for more data is not at the right place. Instead a request to tone down the interpretation is made.

As the first reviewer made a similar point, I include here the answer given there: Unfortunately, the area where A. Matmon and Y. Avni collected the modern chert nodules is very difficult to access these days, and it is not possible to collect any more samples. However, as we point to in the manuscript, there has is an extensive body of work looking into rates of erosion of chert and quartzite surfaces in the hyper-arid Negev desert (e.g., Boroda et al., 2014; Fruchter et al., 2011; Matmon et al., 2009; Matmon et al., 2016; Matmon and Zilberman, 2017). We now references rates from these studies (see lines 307-309).

2. The method section needs to be set-up in a logical and rigorous way (see detailed comments below). The different methods applied need to be described in more detail. The calculations performed and parameters used explained. In general, the order of the presented information could be rearranged to make understanding easier. Concise wording and details in tables and figures are needed (see comments below).

We have somewhat revised the methods section based on comments made here and by the other reviewers. See revisions made to sections 2 and 3.

3. The interpretation of the data needs to be more rigorous. For instance, the nuclide concentrations could be normalized to the same altitude. This would make a comparison more meaningful. A chert nodule at 1000 m above sea level subjected to slow erosion rates is expected to have high nuclide concentrations.

We generally agree with this comment and believe that the data is now presented in a more precise and correct manner. We have now revised the manuscript so we mostly compare exposure times. We still present cosmogenic concentration but solely to examine differences between different samples at the same site (figure 3). Lastly, we present a comparison between exposure times that includes the different possible paleo-erosion rates (see table 2).

Concentrations and erosion rates could also be investigated with "banana plots" for the three different nuclides (e.g., Ivy-Ochs and Kober, 2008). Such plots would help to visualize the data presented in Table 1. In addition, the use of erosion rate in combination with the integration time could be helpful for the reader.

*Unfortunately, applying a 'banana-plot' type of diagram will not help in investigating the reported data. The Miocene sample presented have been buried for extensive periods and so the measured concentrations of cosmogenic 26Al and 10Be do not represent burial time, as the samples have reached a steady state determined by post-burial (muonic) production. We did include a figure that presents changes in steady state concentration with time (see fig. 4).*

Unfortunately, I am not an expert in the measurements of cosmogenic 21Ne. Therefore, the quality and reliability of the presented measurements are not assessed in this review. This is hopefully addressed by another reviewer. For instance, it is not clear to me what happens to 21Ne during sample preparation when leached at 150_C (see line 118-9)? This question comes up as just some lines above (86-9) the closure temperature for 21Ne in quartz is mentioned to be 90 - 100_C. As diffusion is a slow process, the used sample preparation may be valid. Clarification is needed.

*As the manuscript suggests, 80-90°C is the closure temperature of Ne in quartz over geological timescales. The fractional loss of Ne due to diffusion over a 1.5 hour timeframe is insignificant (<0.1%). Nevertheless, we now include a more rigorous examination of possible diffusion from the samples based on comments made by the third reviewer (see lines 106-112 & 193-203).*

**Detailed comments:**

Abstract

L11-14 The abstract jumps to much into details which are not relevant for the abstract. It would make more sense to give the reader a reason why this study was done and what the goal is.

*The abstract has been revised (see lines 25-27). However, we think a short geological background is important for the reader to understand the context of this work and that it is a good starting point.*

L15-18 What do you mean by "modern erosion rates"? Do these rates not integrate over hundred thousand of years? Please clarify.

*This part has been revised. See lines 14-18.*

L16-17 Would it make more sense to first report nuclide concentrations and then move to erosion rates?

*This part has been revised. See lines 14-18.*

L20-21 Is bedrock erosion equal to modern erosion?

*This part has been revised. See lines 21-25.*

L21-23 Long sentence not easy to understand. What does the even mean? Are the currently eroding chert nodules not used for the bedrock erosin? Needs clarification.

This part has been revised. See lines 21-25.

L24-25 From what material are the "rates calculated today"? And what does it mean "rates calculated today"? As mentioned in the general comments, it might be better to compare (normalized) concentrations rather than erosion rates.

This part has been revised. See lines 21-25.

1 Introduction

L37-40 This sentence is not easy to understand for a reader not familiar to the technique of in situ-produced cosmogenic nuclides. Can this sentence be extended? What is the limitation of the method due to the half-lives?

The concept of half-lives is well known in the field of Earth Sciences and we feel that further explanation is beyond the scope of this manuscript. Further information can be found in the reference cited (line 44).

L44 ": : :parts of the: : :"

Corrected. See line 48.

L46-48 The introduction comes to a quick end. An outline of the study set-up and a hypothesis to be tested would be helpful for the reader. What kind of samples do you analyze with what method? What are the questions to be answered with this study?

The introduction has been revised. See lines 50-54.

2 Geological Background

L49 This chapter would gain a lot if called "Study Area" and start with a geographic description of today and what deposits are present. Then move over to the geologic background. Changing this order would request to change Fig. 2 to Fig. 1 and vice versa.

Section 2 has been revised (see lines 56-95) and its name was changed to 'Geological Setting'. However, as we feel that parts of this comment have more to do with writing style choices than substance we choose to keep the name of this section in its current format.

3 Methodology and Analytical Procedures

L76-89 This first section could be labeled 3.1 Cosmogenic method. It could contain the existing information, but also explain the method of catchment-wide erosion rates. In addition, it could contain further explanations of possible problems faced. Not all readers know the caveats of the cosmogenic nuclide methods (e.g., radioactive versus stable nuclides). Would it make sense to discuss here the possible influence of the transport distance and time on radioactive and stable cosmogenic nuclides?

The methods section has been revised (See section 3.3, lines 152-156). Section 3.1 now includes an explanation on Neon-21 accumulation in sedimentary cycles (see lines 98-107). However, as we feel that parts of this comment have more to do with writing style choices than substance we choose to keep the name of this section in its current format.

L90 Now start here: 3.2 Sampling Strategy?

See previous comment.

L134 How is the chemistry blank correction performed? What are the values used for erosion rate calculations?

Blank correction is a commonly performed procedure and further explanation of how exactly it was performed is not in the scope of this manuscript. Regarding erosion rates values see section 3.3, lines 152-156.

4 Results

L143-5 This sentence needs clarification. What is exactly 21Neex?

In addition to the explanation in Table 1, we now include the formula for calculating 21NeEx in the text, see lines 166-167.

L160-1 What is about differences in 21Neex in chert and quartz samples? Could this have an influence on the 21Necos concentration?

We are not sure what the reviewer meant in this comment. However, if the meaning is the differences in cosmogenic 21Ne concentrations (or 21NeEx concentrations) between quartz and chert, those are are presented in Table 1 and explained in the next line below.

L178-9 Difficult sentence to understand. Please clarify.

Al/Be ratios are commonly used for burial ages and as mentioned in the text, this part is thoroughly discussed in the Discussion section and is further explained there.

5 Discussion

L193 What are the parameters used for these calculations?

The parameters are now specified in figure 4 caption and can be found in the referenced sources.

L210 The use of "Modern" is misleading.

This has been revised. See line 245.

L211 On what are the calculations of the erosion rates based? Sea level-high latitude production rate, production rate scaling, etc.? This information should be given in the methods.

We have added the requested details and moved this section to the methods (see section 3.3, lines 152-156).

L215 Are these reported values correct? Please cross-check.

The values used for the calculation were correct, and the mistake made in the text is now corrected (see lines 152-156).

Figures

Fig. S5 - S11: Please label x- and y axes.

Axes labeled. See appendix.

Tables

Table 1: Please cross-check units.

Units have been checked and corrected. See Table 1.

**Response to Reviewer no. 3: Marissa Tremblay**

In this discussion paper, Ben-Israel et al. present new neon isotope measurements from quartz sand and chert pebbles deposited by the Hazeva River, which drained the Arabian Peninsula during Miocene times, as well as from modern eroding outcrops where the chert pebbles were likely sourced. The authors compare apparent erosion rates calculated from cosmogenic 21Ne concentrations at both sites, and conclude that the erosion rates recorded by the Miocene fluvial deposits are higher than those in the modern. They attribute higher Miocene erosion rates to higher uplift rates and a wetter climate of the Arabian Peninsula at that time.

**Major comments:**

In general, I think the approach taken in this paper to quantify paleo-erosion rates is exciting.

We thank Dr. Tremblay for this comment.

However, I am concerned that the uncertainties in the paleo-erosion rates the authors calculate are underestimated, and that therefore the conclusions about higher erosion rates during the Miocene are overstated. Specifically, the authors assume that the elevation during Miocene production of cosmogenic 21Ne was 1000 km. It is unclear to me if this is the assumed elevation of the source of the chert pebbles, and that the authors then assume that the majority of cosmogenic 21Ne production occurs prior to sediment transport? Or is 1000 km accounting for sediment transport and meant to be representative of some catchment-integrated value between where the pebbles were sourced and deposited? Furthermore, it appears that the authors do not give this paleo-elevation any uncertainty in their calculation of exposure times or minimum erosion rates. I suspect that if the authors incorporate a reasonable elevation uncertainty of something like ï´C´s 500 m, that their paleo-erosion rates will overlap entirely with their modern erosion rates. Because the choice of a paleo-elevation has such a large effect on the calculated paleo-erosion rates, there needs to be (1) a more detailed explanation of what the paleo-elevation the authors use represents, and (2) an uncertainty associated with this paleo-elevation incorporated into the calculated paleoerosion rates.

This is a very good point and one that we now incorporate along with other comments regarding paleo-production rates and paleo-elevation of the Miocene sediments. We now consider a range of possible elevations between 500-1000 m.a.s.l and the resulting production rates and exposure times (see Table 2). We have revised the discussion (lines 246-323), conclusions (lines 378-393), and abstract (lines 15-27) accordingly.

I am also wondering if the authors need to be worried about neon diffusion with their analysis chert pebbles. They cite a neon closure temperature for quartz reported by Shuster and Farley (2005), but this was calculated for a 500 micron quartz grain. What is controlling the diffusion lengthscale in these chert pebbles, and what is the typical grain size in the chert pebbles analyzed? If you look at Figure 4 from Shuster and Farley (2005), for a 200 micron-diameter quartz grain (e.g., log radius of -1), you would expect significant diffusive loss of neon on 100 ka timescales and at temperatures of 60-70 certainly be in the range of 60-70 C (e.g., McFadden et al., 2005). Additionally, the fact that the authors observe lower

degassing temperatures in the laboratory for the chert samples than they do for the quartz samples suggest that the chert has a lower thermal sensitivity. Altogether, this makes me think that diffusion might be contributing to the observation that the chert pebbles have lower cosmogenic 21Ne concentrations than the quartz sands. Given this, I think a discussion of the potential role of neon diffusion needs to be added to the text. C. These seem like high temperatures, but in the Arabian Peninsula air temperatures regularly exceed 40 C in the summer months and rock temperatures can certainly be in the range of 60-70 C (e.g., McFadden et al., 2005). Additionally, the fact that the authors observe lower degassing temperatures in the laboratory for the chert samples than they do for the quartz samples suggest that the chert has a lower thermal sensitivity. Altogether, this makes me think that diffusion might be contributing to the observation that the chert pebbles have lower cosmogenic 21Ne concentrations than the quartz sands. Given this, I think a discussion of the potential role of neon diffusion needs to be added to the text.

Two very interesting points are made here regarding (1) the 'typical grain size' in the chert pebbles and (2) the temperatures reached in the Negev Desert. We have revised the manuscript to include the possible role of Ne diffusion, and our answers are as follows:

(1) We do not know with certainty whether or by how much the kinetic parameters of chert differ from that of quartz. This uncertainty and its possible implications are now discussed in lines 193-196.

(2) While it is very true that air temperatures get very high in the desert and the temperatures of exposed dark rock (such as cherts) can get up to even higher reaching 60-70°C and very possibly even higher. However, it is crucial to remember that the Miocene samples presented are fluvial samples that are not likely to be exposed to direct solar radiation for extended periods. In support of this claim, the examined chert samples did not exhibit any visible cracking or fractures commonly identified with thermal stresses (see lines 2198-203).

**Minor comments:**

Figure 1 should have a box indicating the location of figure 2A.

See revisions to figure 2. However, we feel that adding a location box to figure 2 is somewhat redundant, as the major geographical features are marked in both figure 1 and figure 2 so comparison is easy. Furthermore, as the assumed drainage basin of the Hazeva River is marked, it is reasonable not to include additional marking.

Figure 2 should also include field photos of the modern in situ sampling sites of chert nodules.

Unfortunately, these do not exist. While we wish we can go back and take such photos, these sites are no longer accessible.

Lines 204-206: This sentence is awkwardly worded, and I'm not sure I fully understand I follow the logic.

We have rephrased this sentence, hopefully it is now clearer. See lines 238-240.

Lines 255-268: The calculated paleo-erosion rates overlap with the upper end of the modern erosion rates, even without my concerns about the paleo-elevation uncertainty being addressed. This sentence (and similar statements elsewhere) overstates the significance of the authors findings. I think it would be more appropriate to say that the calculated paleo-erosion rates allow for the possibility of higher erosion rates in the Miocene.

This part has been rephrased. See lines 325-335.

Lines 281-283: Here and elsewhere, do you mean rock uplift or surface uplift?

This has been clarified. See lines 345-348.

Supplement: There needs to be some text explaining what is provided in each of the Excel spreadsheet tabs as well as a caption provide for each of the supplemental figures. It's not obvious to me why you need all of the different neon three isotope plots and why they are in the order that they are presented in. This could be cleaned up by having one three isotope plot for the Miocene samples and one for the modern samples, and using different symbol shapes to represent the different temperature steps.

We have made corrections to the supplement based on these comments. See supplementary data.

The revised manuscript with the author's changes included

[revised manuscript text omitted]

---

## Author Response (AR2)

09-Mar-20

Dear Dr. Masteller,

We are very pleased that you as well as both the reviewers are generally pleased with the revised manuscript. We would like to thank both reviewers again, for their hard and detailed work, which have much improved this manuscript. We are pleased to submit are revised manuscript and we believe that it is now suitable for publication.

Regarding the two main suggestions raised:

1) We have revised the discussion section to include the paleoclimate records that were previously introduced in the conclusions. We thank the reviewers for this suggestion.

2) Regarding a more thorough discussion of certain references, I will address each mentioned reference separately.

   - Regarding the Feinstein et al. (2013) paper. In this paper the authors discuss the uplift and denudation history of the eastern Dead Sea rift, SW Jordan, using evidence from fission-track data. In general, we have complied with previous and current comments and have expanded the section in the discussion that addresses the uplift history of the region. However, we would like to remind the reviewers, that while the uplift history of the Suez Rift is interesting it is not the focus of this manuscript. We reference the paper by Feinstein et al. (2013) three times in the manuscript and we do not see any reason to discuss it further, as it does not contribute additional information pertaining to this manuscript.

   - Regarding further discussion of the Sinclair et al. (2019) paper, in this manuscript the authors use cosmogenic 21Ne to study the duration of sediment routing from source to sink. This is an interesting and innovative application of cosmogenic 21Ne, especially coupled with the developed model. While we do reference the manuscript, we believe the data presented in this manuscript is robust and does not warrant additional discussion. A comparison to the Sinclair et al. (2019) paper will neither strengthen nor weaken our interpretation of the data and is therefore beyond the scope of this work.

Regarding the suggested technical corrections, we have complied with the majority of the suggestions as well as fixed incomplete sentences, typos, and grammatical errors throughout the manuscript. Please see the revised manuscript submitted.

We do however oppose the following, small number, of suggestions, see are response in blue:

L339-L344: I think that you should clarify here that chert pebbles were only active during the Miocene, as compare to the recycled sand samples, and that their inheritance of 21Ne from previous cycles inhibits their use to infer erosion rates. In particular, I am not sure the contrast between the exhumation and transport histories of the sand and chert and made as distinct as they could be. I suggest the authors clarify this point.

While we now include additional explanation of this issue (see lines 299-306), we would like to point the reviewer to lines 102-108 in section 3.1 where we go into details regarding this difference between the cherts and quartz. Therefore, we do not think an additional clarification is needed.

Fig. 2a – please indicate units for elevation colorbar

Please see that above the colorbar m.a.s.l is indicated.

Fig 4 – use of a log scale may make the data more visible (especially in panel 3) – you might want to give that a try and see how it looks. I defer to the authors on this.

While we have considered and attempted this suggestion, we believe that this presentation of the data makes the comparison between the radioactive nuclides reaching an equilibrium and the stable nuclide increasing linearly, clearer.

Please find attached below the revised manuscript. Page and line numbers refer to the revised version of the manuscript.

Sincerely,

*Michal Ben-Israel*

Michal Ben-Israel
(on behalf of all the authors)

---

## Author Response (AR3)

**Response to referees - ESurf-2019-54**

Dear prof. Hovius,

Thank you and the associate editor for the editing advice. We have inc comments suggested – see correction in the attached revised manuscript.

Regarding the mentioned citation. We agree with reviewers and the assoc publication (Sinclair et al., 2018; *geology*) is cited (see line 45).

We are happy to submit the final version of the manuscript for publication in

On behalf of all the authors, I would like to thank the entire editorial staff an health during these times.

Sincerely,

Michal Ben-Israel Michal Ben-Israel

**Response to Reviewer no. 1: Taylor Schildgen**

**General comments:**

Ben-Israel and co-authors analyze stable (21Ne) and unstable (10Be, 26Al) of Miocene sediments from the NW Arabian plateau to calculate paleo-erosio "modern" rates obtained from bedrock outcrops. They interpret an appr concentrations to result from erosion rates that were 2x faster during the ea be reasonable considering evidence for a wetter climate in the region at the this work illustrates the unique ability of the stable cosmogenic nuclide 2 averaged over relatively short time intervals (if we can consider 100s of kyr s sedimentary deposits; such information cannot be obtained with 10Be or 26 their consideration of various potential complications of their data – postelevation through time, and how different types of detrital material (quart: have experienced very different pathways to the final deposition site. I also types of detrital material to assess the possibility of 21Ne inheritance.

**We thank prof. Schieldgen for these comments.**

But I see several areas that require improvement. Most concerning for me history constraints, which I don't find very convincing, but are critical for the difference in measured 21Ne concentrations between modern and mexplained either by a change in elevation through time or a change in erosic

This is a very good point and one that has been similarly made by the other this uncertainty in the original manuscript, we agree that the discussion into rates should have been done more circumspectly. We now include additional paleo-elevation and the calculated production rates. In addition to the s evidence we now also calculate the possible elevation using a moderate co to account for the uncertainty in paleo-elevation as well as basin scaled pa consider an elevation uncertainty range of 500-1000 m a s 1 (see lines 250-2 with erosion rates mentioned only in the final part of the manuscript. As m hyper-arid environment, even small changes to erosion rates are significar (lines 333-343) and conclusions (lines 377-381).

I'm also concerned by the small number of samples obtained from the opossibility that the rates reported are not representative of modern rat outcrops lead to a wide range of erosion-rate estimates). Are there any oth have been reported that can be used to corroborate the results present modern erosion-rate estimate wasn't made from modern detrital sands in would not be from exactly the same drainage area as the early-mid Miocen

Unfortunately, the area where A. Matmon and Y. Avni collected the modern to access these days, and it is not possible to collect any more samples. He manuscript, there has is an extensive body of work looking into rates of e surfaces in the hyper-arid Negev desert (e.g., Boroda et al., 2014; Fruchte 2009; Matmon et al., 2016; Matmon and Zilberman, 2017). We now refere (see lines 307-309).

I think the difference between the detrital quartz sand and detrital or emphasized in the final interpretations/conclusions of the paper. It seems the not considered in the final interpretations due to the possibility that the periods of deposition and exposure prior to the last deposition, hence it con the reason that the erosion rates for the quartz samples are not reported, explicitly rather than leaving it for the readers to infer; still I think the erosi But rather than making it seem as if those samples were just a waste or emphasize how in recycled sediments, inherited 21Ne can be a real problem measuring different types of detrital material to assess this possibility! That than hidden.

That was by no means our intention. Inherited cosmogenic 21Ne is one of cosmogenic 21Ne and we include this limitation throughout the manuscript

I. 36-37: Older landscapes are transient? Odd wording. Also, this sentence of previous ones. You've discussed river systems and sediment archives, now landscapes themselves? Be more precise and focused.

We have changed the phrasing in this section. See lines 35-46.

I. 38: Okay, so the focus is on quantifying erosion rates from surfaces? This

**See previous comment.**

I. 41: If the focus is on erosion, don't change the terminology here to encompasses much more than just erosion.

**See previous comment.**

I. 43: Now you've explained that the focus is on sedimentary deposits, r suggest rewriting this whole paragraph with a clearer focus on what inform reader. What is the main problem, why is it difficult to address, how are you

**See previous comment.**

I. 51: Wouldn't it be the other way around, i.e., the Afar plume leads to magr influenced tectonics?

We have changed the phrasing in this section. See lines 56-65.

I. 67-69: This means that the deposition associated with the river started p interpret only the upper part of the Hazeva Formation to be associated with deposition started after 20 Ma? Please clarify.

The sediments in lower part of the Hazeva formation are local and were not (unlike the upper part of the section). We have now clarified this in the text

I. 99-101: How is this history of the quartz sand known? If this history explaining differences between 21Ne measured in quartz vs. chert, then a

This paragraph has been moved and modified. See lines 75-95.

I. 109: How deeply shielded were the collected samples? Deeply enough 21Ne production?

This question is thoroughly discussed in the discussion section. We now ref 116).

I. 113: I suggest "accumulated cosmogenic nuclides only during exhum experienced the full sedimentary cycle also accumulated nuclides during ex

We accept this correction. See line 127.

I. 156-157: Is this because you assume the U and Th are equally distributed reasonable assumption?

U and Th are most likely found in inclusions within the crystal lattice. The U/ is determined by the age of the rock and the environmental conditions metamorphosis). It is reasonable to assume that is would the same or similar as they all share the same lithology.

I. 212-217: Don't assume that your readers remember that EJC5 and EJC3 from the "in situ" outcrops, remind us.

Maybe this comment can be clarified as the first part of the paragraph de 246-259).

I. 214-215: This detail concerning the scaling of production rates belor discussion.

We now include these details in the methods section (see section 3.3, lines

I. 223-224: I don't see the added value of reporting equivalent exposure time "simple exposure time"), given that you are mainly interpreting the measure erosion rates. Or is the goal to give readers a sense of the averaging times

I. 237-239: A bigger overview map that includes the Suez rift in addition to t mentioned here would be very helpful.

**See our revisions to figure 2, and our comments to this there.**

I. 237-242: These uplift constraints are crucial for your interpretation of whe samples show a faster erosion rate compared to today or reflect a similar ero paleo-production rate. Given their importance, some more details on these very helpful. Although I have not checked each of the references in detail, referenced the Wilson et al. (2014) interpretations. Despite many reasons uplift histories from river profiles should be considered suspect, their interpretation most of the modern elevation gain occurred since 20 Ma, and it looks since 10 Ma (see their Fig. 17). For that reason, I don't agree at all with your stop resume that the western flank of the Arabian Peninsula (or the NW edg area) reached its current elevation prior to the initiation of the Hazeva fluvi

As is mentioned in our answers to the general comments, we agree with a regarding the elevation of the Arabian Plateau during the Miocene should h now provide additional evidence to support our assumption of the paleo-e significant uncertainty for this (see lines 250-289). Regarding Wilson et al. (2) figure 21, the central part of the Arabian plate appears to be stable ~20 Myr tip).

I. 255-258: I can guess why you do not mention erosion rates from the quart it has inherited 21Ne – but it seems like an oversight. I suggest to not "h emphasize how recycling of quartz sand can lead to incorrect results.

As is mentioned in our answers to the general comments, hiding this aspect We refer to the possible inheritance in quartz throughout the manuscrip section). The discussion about the possible effects of it is slightly beyond the is referenced to (see Ben-Israel et al., 2018 for further reading).

I. 281-287: Mostly I've been able to work out myself whether you are reference uplift up until now, but in this section in particular I cannot follow your mode and the manuscript specify which one you are referring to

paleo-elevation, still seem very slow. How do rates of 4 to 12 mm/kyr measured from similar environments today? (Incidentally, I realize I'm assured is relatively low, but it would be helpful to actually show a slope/relief map

that's the case).

This section has now been revised - see out previous comments and lines 3

Figure 2: As mentioned above, a broader overview map would be very helpf the bottom? Highlighting or circling him/her in some way would make it eas this photo. Likewise, in 2C, is that a dog?

Figure 2 has been revised and now includes a more zoomed overview map additionally now includes clear marking of a human (Dr. Avni) in 2B and dog

Figure 3: Given the overall focus on erosion rates, I find it odd that the cal shown in this figure. Why not use those instead of the effective exposure as

With the addition of Table 2, figure 3 no longer includes exposure ages (or e

**Editorial comments:**

I. 35: always specify what you mean after "this", e.g., this lack of informatio

Corrected. See line 38.

I. 64: "comprise" rather than "compose"

Corrected. See line 78.

I. 71: I'd suggest "disruption" rather than "dismantlement"

Corrected. See line 71.

I. 210: Please refer to "denudation" or "erosion" rates throughout, not "rate is unnecessarily vague.

We have corrected this in the referenced line (245) and throughout the ma

I. 221: lots of needless words here, please shorten to "erosion rates betwee

This section has been revised. See lines 307-309.

**Response to Reviewer no. 2: Anonymous Referee**

Ben-Israel et al. present 10 new in situ-produced 21Ne concentrations from the pre-Dead Sea rift Hazewa River located in southern Israel. Where possil 26Al concentrations are provided for the same sample material. The data from is used to determine Early-mid Miocene erosion rates.

**General comments:**

The manuscript is generally well written and reads well. However, there a manuscript which need to be clarified and improved:

**We thank the reviwer for these comments.**

1. The interpretations of the data are relatively strong given the amount manuscript stands right now, it is not clear to me if the given interpretation more data is available. For instance, the nuclide concentrations of the two with continuous erosion of a landscape. In order to investigate the problem analyzed. However, knowing that this is easy to say and that cosmogenic n a request for more data is not at the right place. Instead a request to tor made.

As the first reviewer made a similar point, I include here the answer given t where A. Matmon and Y. Avni collected the modern chert nodules is very and it is not possible to collect any more samples. However, as we point to i an extensive body of work looking into rates of erosion of chert and quartz Negev desert (e.g., Boroda et al., 2014; Fruchter et al., 2011; Matmon et al Matmon and Zilberman, 2017). We now references rates from these studie

2. The method section needs to be set-up in a logical and rigorous way (see The different methods applied need to be described in more detail. The parameters used explained. In general, the order of the presented inform make understanding easier. Concise wording and details in tables and figure Concentrations and erosion rates could also be investigated with "banana nuclides (e.g., Ivy-Ochs and Kober, 2008). Such plots would help to visualiz 1. In addition, the use of erosion rate in combination with the integration reader.

Unfortunately, applying a 'banana-plot' type of diagram will not help in inv The Miocene sample presented have been buried for extensive per concentrations of cosmogenic 26Al and 10Be do not represent burial time, a steady state determined by post-burial (muonic) production. We did in changes in steady state concentration with time (see fig. 4).

Unfortunately, I am not an expert in the measurements of cosmogenic 21N reliability of the presented measurements are not assessed in this review. The another reviewer. For instance, it is not clear to me what happens to 21N when leached at 150\_C (see line 118-9)? This question comes up as just closure temperature for 21Ne in quartz is mentioned to be 90 - 100\_C. As a used sample preparation may be valid. Clarification is needed.

As the manuscript suggests, 80-90°C is the closure temperature of Ne in qua The fractional loss of Ne due to diffusion over a 1.5 hour timeframe is insign we now include a more rigorous examination of possible diffusion from the made by the third reviewer (see lines 106-112 & 193-203).

**Detailed comments:**

**Abstract**

L11-14 The abstract jumps to much into details which are not relevant for more sense to give the reader a reason why this study was done and what t

The abstract has been revised (see lines 25-27). However, we think a sh

L21-23 Long sentence not easy to understand. What does the even mean? A nodules not used for the bedrock erosin? Needs clarification.

This part has been revised. See lines 21-25.

L24-25 From what material are the "rates calculated today"? And what do today"? As mentioned in the general comments, it might be better concentrations rather than erosion rates.

This part has been revised. See lines 21-25.

**1 Introduction**

L37-40 This sentence is not easy to understand for a reader not familiar produced cosmogenic nuclides. Can this sentence be extended? What is the to the half-lives?

The concept of half-lives is well known in the field of Earth Sciences and we is beyond the scope of this manuscript. Further information can be found in

L44 ": : : parts of the: : :"

Corrected. See line 48.

L46-48 The introduction comes to a quick end. An outline of the study set-up would be helpful for the reader. What kind of samples do you analyze with questions to be answered with this study?

The introduction has been revised. See lines 50-54.

**2 Geological Background**

L49 This chapter would gain a lot if called "Study Area" and start with a ge and what deposits are present. Then move over to the geologic backgrour request to change Fig. 2 to Fig. 1 and vice versa. The methods section has been revised (See section 3.3, lines 152-156). explanation on Neon-21 accumulation in sedimentary cycles (see lines 98-1 parts of this comment have more to do with writing style choices than substance of this section in its current format.

L90 Now start here: 3.2 Sampling Strategy?

See previous comment.

L134 How is the chemistry blank correction performed? What are the calculations?

Blank correction is a commonly performed procedure and further explan performed is not in the scope of this manuscript. Regarding erosion rates val 156.

**4 Results**

L143-5 This sentence needs clarification. What is exactly 21Neex?

In addition to the explanation in Table 1, we now include the formula for o see lines 166-167.

L160-1 What is about differences in 21Neex in chert and quartz samples? Co the 21Necos concentration?

We are not sure what the reviewer meant in this comment. However, if the cosmogenic 21Ne concentrations (or 21NeEx concentrations) between quipresented in Table 1 and explained in the next line below.

L178-9 Difficult sentence to understand. Please clarify.

Al/Be ratios are commonly used for burial ages and as mentioned in the discussed in the Discussion section and is further explained there.

L215 Are these reported values correct? Please cross-check.

The values used for the calculation were correct, and the mistake made in t lines 152-156).

**Figures**

Fig. S5 - S11: Please label x- and y axes.

Axes labeled. See appendix.

Tables

Table 1: Please cross-check units.

Units have been checked and corrected. See Table 1.

**Response to Reviewer no. 3: Marissa Tremblay**

In this discussion paper, Ben-Israel et al. present new neon isotope measur chert pebbles deposited by the Hazeva River, which drained the Arabian Per as well as from modern eroding outcrops where the chert pebbles were compare apparent erosion rates calculated from cosmogenic 21Ne conce conclude that the erosion rates recorded by the Miocene fluvial deposits modern. They attribute higher Miocene erosion rates to higher uplift rates Arabian Peninsula at that time.

**Major comments:**

In general, I think the approach taken in this paper to quantify paleo-erosio

**We thank Dr. Tremblay for this comment.**

However, I am concerned that the uncertainties in the paleo-erosion raunderestimated, and that therefore the conclusions about higher erosion overstated. Specifically, the authors assume that the elevation during Mioce 21Ne was 1000 km. It is unclear to me if this is the assumed elevation of the and that the authors then assume that the majority of cosmogenic 21N sediment transport? Or is 1000 km accounting for sediment transport and r some catchment-integrated value between where the pebbles wer Furthermore, it appears that the authors do not give this paleo-elevat calculation of exposure times or minimum erosion rates. I suspect that reasonable elevation uncertainty of something like i'C 's 500 m, that their pa entirely with their modern erosion rates. Because the choice of a paleo-elevat paleo-elevation the authors use represents, and (2) an uncertainty associal incorporated into the calculated paleoerosion rates.

**This is a very good point and one that we now incorporate along with othe**

degassing temperatures in the laboratory for the chert samples than the suggest that the chert has a lower thermal sensitivity. Altogether, this makes be contributing to the observation that the chert pebbles have lower cosr than the quartz sands. Given this, I think a discussion of the potential role added to the text. C. These seem like high temperatures, but in the Arabia regularly exceed 40 C in the summer months and rock temperatures can car 70 C (e.g., McFadden et al., 2005). Additionally, the fact that the auth temperatures in the laboratory for the chert samples than they do for the que chert has a lower thermal sensitivity. Altogether, this makes me think that d to the observation that the chert pebbles have lower cosmogenic 21Ne cos sands. Given this, I think a discussion of the potential role of neon diffusion

Two very interesting points are made here regarding (1) the 'typical grain (2) the temperatures reached in the Negev Desert. We have revised the man role of Ne diffusion, and our answers are as follows:

(1) We do not know with certainty whether or by how much the kinetic pathat of quartz. This uncertainty and its possible implications are now discus

(2) While it is very true that air temperatures get very high in the desert and dark rock (such as cherts) can get up to even higher reaching 60-70°C ar However, it is crucial to remember that the Miocene samples presented ar likely to be exposed to direct solar radiation for extended periods. In suppor chert samples did not exhibit any visible cracking or fractures commonly ide (see lines 2198-203).

**Minor comments:**

Figure 1 should have a box indicating the location of figure 2A.

Lines 255-268: The calculated paleo-erosion rates overlap with the upper end even without my concerns about the paleo-elevation uncertainty being a similar statements elsewhere) overstates the significance of the authors find appropriate to say that the calculated paleo-erosion rates allow for the pos in the Miocene.

**This part has been rephrased. See lines 325-335.**

Lines 281-283: Here and elsewhere, do you mean rock uplift or surface upli

This has been clarified. See lines 345-348.

Supplement: There needs to be some text explaining what is provided in e tabs as well as a caption provide for each of the supplemental figures. It's need all of the different neon three isotope plots and why they are in the in. This could be cleaned up by having one three isotope plot for the Mioo modern samples, and using different symbol shapes to represent the different

We have made corrections to the supplement based on these comments. S

**The revised manuscript with the author's changes included Early-mid Miocene erosion rates measured in Dead Sea rift Hazeva River fluvial cher cosmogenic 21Ne-in fluvial chert pebbles**

- 5 Michal Ben-Israel1, Ari Matmon1, Alan J. Hidy2, Yoav Avni3, Greg B.
  1The Institute of Earth Sciences, Hebrew University of Jerusalem, Jerusalem, 91904
  2Center for Accelerator Mass Spectrometry, Lawrence Livermore National Laborate
  3Geological Survey of Israel, Yesha'yahu Leibowitz 32, Jerusalem, 96921 Israel
  4Berkeley Geochronology Center, Berkeley, California 94709, USA
- 10 Correspondence to: Michal Ben-Israel (michal.benisrael@mail.huji.ac.il)

15

Abstract. The Miocene Hazeva River was a large fluvial system >100,000 km2) that drained the Arabian Plateau and Sinai Peninsula during the Early-Mid Miocene. It was established after rifting of the R Plateau during the Oligocene. Following late Miocene to early Pliocene Sea Rift, the Hazeva drainage system was abandoned and dissected

divides-on either side of the rift. In this work, Wwe utilized a novel

Rift, the Hazeva drainage system was abandoned and dissected, result

25 on either side of the rift. We find that modern erosion rates derived fra and 10Be in exposed *in situ* chert nodules to be extremely slow, between between modern and paleo-paleo-erosion rates, measured in chert peb as cosmogenic 21Ne was acquired partly during bedrock exhumation transport of these pebbles in the Hazeva River. However, even
30 maintained transport along this big river, 21Ne concentrations exposure in Miocene cherts are generally lower shorter (range between 0+5/0 kyr.97±1.39×106 atoms/g-SiO2) compared to exposure times calculated in the currently eroding chert nodules presented here

 $378\pm762.10\pm2.43\times10^{6}$  atoms/g SiO2kyr) and other chert surfaces cur

35 environments. Shorter\_21Ne concentrations exposure times in Miocene minimum-paleo-erosion rates, that are at least twice as fast as rates er attribute these faster erosion rates to a combination of continuous sur wetter climatic conditions during the early-mid Miocene.

**1. Introduction**

40 Tectonic and climatic conditions control geomorphological processes

50 climatic conditions that prevailed. This lack of information is preservation potential of older sediments landscapes in fluvial systems these tend to end up either deeply buried at depositional basins or reprocess (e.g., Anderson et al., 1996; Guralnik et al., 2011; Schaller et of transient landscapes. Cosmogenic nuclides have long been applie 55 diverse geological settings (e.g., Bierman, 1994; von Blanckenburg when geological circumstances do allow for the preservation of sk sediments, erosion rates prior to the Pliocene cannot be quantified wit cosmogenic radionuclides (10Be and 26Al) due to their half-lives accordingly; Ivy-Ochs and Kober, 2008). Unlike their radioac cosmogenic nuclides have the potential to quantify rates of surface pro 60 Cretaceous - of surface processes significantly older than cor radionuclides (Balco et al., 2019; Ben-Israel et al., 2018; Dunai et al., Sinclair et al., 2019). Here, we apply stable cosmogenic 21Ne to sedi early-mid Miocene (~18 Ma) by a massive fluvial system that dra Peninsula and Sinai into the Mediterranean prior to the subsidence of 65 Dead Sea transform (Garfunkel and Horowitz, 1966; Zilberman and the time of exposure during erosion and transport of Miocene cher Hazeva River and compare it to exposure times of chert that has been  $(\sim 10^5 \text{ yr})$ . Through this comparison, we compare The erosion rates of fromduring early-mid -Miocene to those measured today in hyper-arid 70 examine the possible influence of river sediments open a window inf

- 80 of the Afar plume uplifted the Arabian Peninsula has been uplifting present elevation of ~1km (e.g., Feinstein et al., 2013; Morag et al., 20 a result of this uplift, widespread <del>erosion denudation followed, and foll</del> truncation surface developed in the northern Red Sea and the south older strata down to Precambrian formations depending on the preext
- 85 2012). Following these events, Dduring the early-mid Miocene, the up a newly established fluvial system, termed the Hazeva River, which f the uplifted eroded terrains towards the Mediterranean Sea, and >100,000 km2 (Garfunkel and Horowitz, 1966; Zilberman and Calvo, fluvial system operated until the subsidence of the Dead Sea Rift, during the subsidence of the Dea
- Pliocene, brought on a dramatic change in morphology, which led to the fluvial system, the last of its kind in the region (Garfunkel, 1981).
   Hazeva River was abandoned, and new independent drainage system the Dead Sea Basin (Avni et al., 2001).

At present, the mostly clastic sedimentary Miocene sequence deposite

- 95 systemRiver is preserved mainly in structural lows, karstic systems, an in southern Israel, eastern Sinai, and Jordan (Calvo and Bartov, 20) associated with this Miocene fluvial system compose comprise the u formation in southern Israel. This formation is divided into two majo autochthonous conglomerates and lacustrine carbonate units, and the
- 100 allochthonous clastic sequences typical to fluvial environments (Calvo sand and chert pebbles . Here we focus on the allochthonous upper pa

onset of the Hazeva River is constrained by the Karak dike (~20 My section of the Hazeva formation (Calvo and Bartov, 2001). While clima during the Miocene are believed to have been wetter (e.g., Kolodny region is part of a middle latitude dry warm desert extending from north

115 with the Negev Desert remaining hyperarid at least since the middle 2006). The Hazeva fluvial system operated until the subsidence of the late Miocene to early Pliocene brought on a dramatic change in modismantlement of this massive fluvial system, the last of its kind in the By the early Pliocene, the Hazeva River was abandoned, and new incoderation drained the region toward the Dead Sea Basin (Avni et al., 2001).

3. Methodology and Analytical Procedures

**3.1 Sampling Strategy**

Cosmogenic nuclides in sediments accumulate throughout the sedime material is exposed during weathering and exhumation exposure of transport in a specific drainage system, and to a much lesser degre intermediate or final destination. Unlike the more commonly use nuclides, which may decay substantially or even completely over m 21Ne is stable. This means that the concentration of 21Ne measured accumulated over several sedimentary cycles of exposure and deposi

130 reaches the depositional basin, it can be re-exhumed and once again e new sedimentary cycle. Therefore, the concentration of cosmogenic obtained from two Miocene Hazeva deposits exposures (Fig. 2 B-0 samples were collected from deeply shielded locations to minimize production (see section 5.1 for further discussion). The quartz sand a both transported by the Miocene Hazeva system and share an overa

- 145 However, the quartz sand was exposed in previous sedimentary cycle and Paleozoic, where it accumulated cosmogenic 21Ne. In contrast, the Eocene and then exposed, transported, and buried during the Mid Therefore, while the cosmogenic 21Ne measured in the quartz sand repricycles, the cosmogenic 21Ne measured in the chert pebbles represents ended.
- a single sedimentary cycle in the Miocene Hazeva River.
   Additionally, Ttwo individual samples of *in situ* chert nodules (EJC3 from exposed bedrock outcrops of the Eocene source rock in central J Miocene samples, which were exposed during at least one full sedim chert nodules accumulated cosmogenic nuclides only during exhumati surface. Therefore, the cosmogenic nuclide These-concentrations meas thus represent averaged rates of erosion surface denudation over the laboration.

**3.2 Preparation of Chert and Quartz Samples and Analytical Proc**

Chert pebbles (ranging 4-14 cm, b axis) were crushed and both chert art to 250-850 µm. Chert and quartz samples were processed to separate of

160 Earth Sciences Cosmogenic Isotope Laboratory, Hebrew Universit standard procedures (Hetzel et al., 2002; Kohl and Nishiizumi, 199 leached in HCl/HNO2 mixture (3:1) at a temperature of 150°C for 1.51 each temperature step. Ne isotope measurements used the BGC "Ohio described in Balco et al., (2019). 20-30 grams of leached and clea samples and three chert samples were processed to separate Be and Al

175 Nishiizumi (1992) and Bierman and Caffee (2001). These were then 26Al/27Al at the Centre for Accelerator Mass Spectrometry, Law Laboratory, and calibrated against house standards and blanks.

**3.3 Cosmogenic Scaling and Correction Factors**

Exposure and burial times and erosion rates were calculated based of

180 using time-independent scaling (Stone, 2000) and production mechan (2008), given sea-level high-latitude production rates of 4.96 atoms atoms/g SiO2/year for 26Al (Balco et al., 2008), and Exposure ages, rejusing production rates scaled for latitude and altitude after Stone (20 rate of 18.1 atoms/g SiO2 year (Borchers et al., 2016; Luna et al., 2018)

**185 **4. Results**

**4.121Ne in Quartz Sand and Cherts**

For the chert samples, <2% of the total 21Ne and no more than 1% of the released above 950°C (see the Supplementary Tables S1-4). Therefore performed at 450, 700, and 950°C heating steps for chert samples and

190 steps for quartz samples (Table 1). Of the total 21Ne measured, >85o temperature steps, below the 950°C step in the chert samples and be quartz samples (see Supplementary Tables S1-4). Also, low-temperature calculations, discussion, and interpretations. It is important to note isotopic values of  ${}^{21}$ Ne/ ${}^{20}$ Ne and  ${}^{22}$ Ne/ ${}^{20}$ Ne ratios at the low-temperatu cosmogenic component of  ${}^{21}$ Neex from the nucleogenic component, p and Th within the crystal lattice, is not trivial. Nonetheless, as all cl

nodules and Miocene chert pebbles) share the same lithology, any concentrations must be due to the cosmogenic component. The chert pebbles and quartz sands sampled at both Miocene H

205

concentrations of 21Necos ranging between 0.00±1.88·106 and 8.89±1.8

- 210 At both Miocene Hazeva sites, the cosmogenic 21Ne concentrations m similar or lower compared to sand samples. These measured con understanding that the sand samples contain quartz grains that originat that were deposited throughout the Phanerozoic and could have und cycles before they were exhumed and transported by the Miocene flu
- the sand samples could have higher concentrations of nucleogenic 21N sand is >800 Ma (Kolodner et al., 2009). Conversely, the chert sa relatively young, Eocene, source rock and only participated in one se Miocene. Both chert nodule samples collected from *in situ* Eoce cosmogenic 21Ne concentrations, higher compared to the Miocene che
  Diffusion kinetics of Ne in quartz have been examined experimentally and Farley, 2005; Tremblay et al., 2014) but have yet to be tested o unclear what is the diffusion length-scale of chert crystals. While dif likely to be similar to quartz, more work is needed to determine that we have the second se

10Be and 26Al concentrations were measured in three Miocene sand sa MHS5), the two Eocene chert nodules (EJC3 and EJC5) and two c MHC6). 10Be results for sample MHC5b and 26Al results for sample MH

- 1). Miocene sand and chert samples show 10Be and 26Al concentrations with extended periods of burial (≤0.39.±0.03·105 atoms/g SiO2 for atoms/g SiO2 for 26Al). Currently eroding Eocene nodules show higher 26Al, with sample EJC3 showing 26Al/10Be ratio that is consistent with and sample EJC5 showing a lower 26Al/10Be ratio, suggesting a more co (see Discussion section).
  - 5. Discussion

5.1 Correcting for Post-Burial Muonic Produced Cosmogenic 21Ne When examining concentrations of cosmogenic nuclides in sediment extended periods, post-burial production needs to be considered. At or
interactions are the main pathway for *in situ* production of cosmogenic >95% for 26Al, 10Be, and 21Ne (Dunai, 2010). However, the relative comuon interactions increases with burial depth, and while production racan be significant when integrated over long periods of time—especial post-burial component does not represent surface processes, and there

250 for its contribution to the measured cosmogenic component. For radioa such as 10Be and 26Al, their initial concentrations (acquired during exp due to radioactive decay, with 26Al decreasing faster than 10Be accor

burial depth, only tens of meters below the surface, and the deduced bu of surface erosion that occurred during the last ~2 Myr (Matmon references therein). Additionally, the relatively large uncertainty on could account for some of this discrepancy (Balco, 2017; Balco et a show that the cosmogenic 21Ne produced post-burial over 18 Myr of b 120 m is lower than the 21Neex measured for in the presented samples (in ). The maximal calculated post-burial cosmogenic 21Ne concentration is lower than the 31O23. This which concentration is lower than the 31O23.

270 all measured Miocene samples except for MHC2, where no cosmo However, sample MHC2 is not considered in the interpretations of consider post-burial cosmogenic 21Ne production to be insignificant exposure times.

**5.2 Calculating Modern and Miocene Exposure TimesRates of Sur**

- 275 Exposure times Erosion rates calculated for exposure at the surface concentrations measured in modern exposed in situ chert nodules for Plateau (EJC3 and EJC5) range between a minimum of 193 kyr and a minimum of 193 kyr and a minimum of 193 kyr and a minimum of 8.08±1.48·106 and SiO2).
- 280 Erosion rates calculated from 10Be and 26Al concentration measured 2-4 mm/kyr, with production rates scaled for latitude and altitude production rates of 2.62 and 30.26 atoms/g SiO2 year for 10Be and 26/ erosion rates calculated from 10Be and 26Al concentrations measured

In comparison to the Jordanian samples, Qquantifying\_rates of surface samples exposed\_that occurred during the Miocene using cosmogenia

[revised manuscript text omitted]

Carrier | 10 Be/ 9 Be | [ 10 Be]                             |  |
|--------|----------------|------------------------|------------------------------------|----------|-----------|-----------|---------------|-----------------------------------|-------------------------------------------------|--|
|        |                |                        | (m)                                | Lat (°N) | Long (°E) | (m.a.s.l) | (mg)          | (×10 -13 )             | (10 5 atoms/
g SiO 2 ) |  |
| MHS1   | Quartz         | Paran Valley,          | 30                                 | 30.33296 | 34.92724  | 290       | 176           | 0.17±0.03                         | $0.14\pm0.02$                                   |  |
|        | sand           | Israel                 |                                    |          |           |           |               |                                   |                                                 |  |
| MHS3   | Quartz
sand | Arad Quarry,
Israel | 90                                 | 31.23372 | 35.20685  | 570       | 171           | 0.36±0.02                         | 0.29±0.02                                       |  |
| MHS5   | Quartz
sand | Arad Quarry,
Israel | 100                                | 31.23372 | 35.20685  | 570       | 175           | 0.32±0.02                         | 0.26±0.02                                       |  |
| MHC2   | Chert          | Paran Valley,          | 20                                 | 30.33296 | 34.92724  | 290       | NA            | NA                                | NA                                              |  |
|        | pebble         | Israel                 |                                    |          |           |           |               |                                   |                                                 |  |
| MHC3   | Chert          | Arad Quarry,           | 90                                 | 31.23372 | 35.20685  | 570       | NA            | NA                                | NA                                              |  |
|        | pebble         | Israel                 |                                    |          |           |           |               |                                   |                                                 |  |
| MHC5a  | Chert          | Arad Quarry,           | 100                                | 31.23372 | 35.20685  | 570       | NA            | NA                                | NA                                              |  |
|        | pebble         | Israel                 |                                    |          |           |           |               |                                   |                                                 |  |
| MHC5b  | Chert          | Arad Quarry,           | 100                                | 31.23372 | 35.20685  | 570       | 172           | NA                                | NA                                              |  |
|        | pebble         | Israel                 |                                    |          |           |           |               |                                   |                                                 |  |
| MHC6   | Chert          | Paran Valley,          | 30                                 | 30.33296 | 34.92724  | 290       | 170           | $0.10\pm0.01$                     | $0.39 \pm 0.03$                                 |  |
|        | pebble         | Israel                 |                                    |          |           |           |               |                                   |                                                 |  |
| EJC3   | In situ        | Central                | Surface                            | 30.97045 | 36.64469  | 910       | 172           | $0.70\pm0.03$                     | $1.13\pm0.05$                                   |  |
|        | chert          | Jordanian              |                                    |          |           |           |               |                                   |                                                 |  |
|        |                | Plateau                |                                    |          |           |           |               |                                   |                                                 |  |
| EJC5   | In situ        | Central                | Surface                            | 30.87181 | 36.52129  | 1000      | 178           | $18.43 \pm 0.30$                  | $29.75 \pm 0.49$                                |  |
|        | chert          | Jordanian              |                                    |          |           |           |               |                                   |                                                 |  |
|        |                | Plateau                |                                    |          |           |           |               |                                   |                                                 |  |

Table 1: Sample Description, Sampling Site Locations and Cosmogenic Nuc

Note: NA - not available. Samples were either not analyzed, or no result was attained.

\*Measurement uncertainties are ~5%.

†Cosmogenic 21Ne is the excess of 21Ne concentrations relative to the atmospheric 21Ne/20Ne ratio, calculated for the low-temperature steps (<

| Sample | Sample type          | Location                               | Exposure time               | Erosion |
|--------|----------------------|----------------------------------------|-----------------------------|---------|
|        |                      |                                        | (kyr)                       | (mm/kyı |
| MHS1   | Miocene quartz sand  | Paran Valley, Southern Negev Desert    | $114 \pm 46 - 166 \pm 87$   | -       |
| MHS3   | Miocene quartz sand  | Arad Quarry, Northeastern Negev Desert | $280{\pm}10-408{\pm}63$     | -       |
| MHS5   | Miocene quartz sand  | Arad Quarry, Northeastern Negev Desert | $278 \pm 17 - 404 \pm 83$   | -       |
| MHC3   | Miocene chert pebble | Arad Quarry, Northeastern Negev Desert | 167±53 - 242±113            | 3.0±1.4 |
| MHC5a  | Miocene chert pebble | Arad Quarry, Northeastern Negev Desert | $91{\pm}46 - 132{\pm}78$    | 5.5±3.3 |
| MHC5b  | Miocene chert pebble | Arad Quarry, Northeastern Negev Desert | $0^{+59}_{-0}-0^{+85}_{-0}$ | >8.6->  |
| MHC6   | Miocene chert pebble | Paran Valley, Southern Negev Desert    | $121 \pm 59 - 176 \pm 102$  | 3.0±1.4 |
| EJC3*  | In situ chert nodule | Central Jordanian Plateau              | 269±49 / 16±1 / 13±1        | 2.7±0.5 |
| EJC5*  | In situ chert nodule | Central Jordanian Plateau              | 378±76 / 361±6 / 378±3      | 1.9±0.4 |

Table 2: Exposure times and erosion rates calculated for the modern and Miocene samples

*Note:* Exposure times is the 'simple exposure time' calculated for exposure at the surface, calculated cosmogenic 21Ne production rates ranging yr), given an elevation of 500 and 1000 meters above sea level. Erosion rates for sand samples were not calculated as the concentration of cosmic inherited cosmogenic 21Ne from previous sedimentary cycles. \*Erosion rates calculated using 21Ne / 10Be / 26Al.

Figure 1. Paleo-geographic map of the eastern Levant during the ear Meulenkamp and Sissingh, 2003) with the approximated extent of the H on Avni et al., 2012; Zilberman and Calvo, 2013).